# Posttranslational insertion of small membrane proteins by the bacterial signal recognition particle

Ruth Steinberg[1][⍟], Andrea Origi[1,2][⍟], Ana Natriashvili[1,2][⍟], Pinku Sarmah[1,2], Mariya Licheva[1,2], Princess M. Walker[3], Claudine Kraft[1], Stephen High[4], Joen Luirink[5], Wei. Q. Shi[3], Martin Helmstädter[6], Maximilian H. Ulbrich[6,7], Hans-Georg Koch[1]*

1 Institute of Biochemistry and Molecular Biology, ZBMZ, Faculty of Medicine, Albert-Ludwigs-University Freiburg, Freiburg, Germany, 2 Faculty of Biology, Albert-Ludwigs-University Freiburg, Freiburg, Germany, 3 Department of Chemistry, Ball State University, Muncie, Indiana, United States of America, 4 School of Biological Sciences, University of Manchester, Manchester, United Kingdom, 5 Molecular Microbiology, AIMMS, Vrije Universiteit Amsterdam, Amsterdam, the Netherlands, 6 Internal Medicine IV, Department of Medicine, Medical Center − University of Freiburg, Faculty of Medicine, University of Freiburg, Freiburg, Germany, 7 BIOSS Centre for Biological Signalling Studies, University of Freiburg, Freiburg, Germany

⍟ These authors contributed equally to this work.
* Hans-Georg.Koch@biochemie.uni-freiburg.de

**Data Availability Statement:** All relevant data are within the paper and its supporting information files.

## Abstract

Small membrane proteins represent a largely unexplored yet abundant class of proteins in pro- and eukaryotes. They essentially consist of a single transmembrane domain and are associated with stress response mechanisms in bacteria. How these proteins are inserted into the bacterial membrane is unknown. Our study revealed that in *Escherichia coli*, the 27-amino-acid-long model protein YohP is recognized by the signal recognition particle (SRP), as indicated by in vivo and in vitro site-directed cross-linking. Cross-links to SRP were also observed for a second small membrane protein, the 33-amino-acid-long YkgR. However, in contrast to the canonical cotranslational recognition by SRP, SRP was found to bind to YohP posttranslationally. In vitro protein transport assays in the presence of a SecY inhibitor and proteoliposome studies demonstrated that SRP and its receptor FtsY are essential for the posttranslational membrane insertion of YohP by either the SecYEG translocon or by the YidC insertase. Furthermore, our data showed that the *yohP* mRNA localized preferentially and translation-independently to the bacterial membrane in vivo. In summary, our data revealed that YohP engages an unique SRP-dependent posttranslational insertion pathway that is likely preceded by an mRNA targeting step. This further highlights the enormous plasticity of bacterial protein transport machineries.

The majority of proteins destined for the inner membrane, the periplasmic space, or the outer membrane in bacteria engage two distinct targeting pathways. Inner membrane proteins are cotranslationally recognized by the signal recognition particle (SRP), which in *E. coli* consists of the protein Ffh and the 4.5S RNA [1]. SRP targets ribosome-associated nascent membrane proteins (ribosome-associated nascent chains [RNCs]) to the membrane-bound SRP receptor FtsY. FtsY is bound to either the SecYEG translocon or the YidC insertase, and this close

**Funding:** This work was supported by the German Science Foundation (grants SPP2002/KO2184/9-1; KO2184/8 and Project ID 235777276/GRK2202 to H-GK; Project ID 403222702/SFB 1381 to H-GK and CK; Project ID 259130777/SFB 1177, Project ID 409673687, and Project ID 390939984/CIBSS-EXC-2189 to CK), by the National Institute of General Medical Sciences of the National Institutes of Health (NIH) (grant no. R15GM116032) to WQS, by the National Science Foundation LSAMP Program (grant no. HRD 1618408) to PMW and by an FF-Nord Fellowship to RS. SH is supported by a Welcome Trust Investigator Award in Science 204957/Z/16/Z. The Kraft laboratory has received funding from the European Research Council (ERC) under the European Union's Horizon 2020 research and innovation program under grant agreement no. 769065, from the European Union's Horizon 2020 research and innovation program under grant agreement no. 765912, and from the EMBO Young Investigator Program. The funders had no role in study design, data collection and analysis, decision to publish, or preparation of the manuscript.

**Competing interests:** The authors have declared that no competing interests exist.

**Abbreviations:** AMP, antimicrobial peptide; CTF, cytosolic translation factor; GFP, green fluorescent protein; INV(OE, INV from a SecYEG-overproducing *E. coli* strain; INV, inner membrane vesicle; IpomF, Ipomoeassin F; IPTG, isopropyl 1-thio-β-D-galactopyranoside; MtlA, mannitol permease; OM, outer membrane fraction; pBpa, para-benzoyl-L-phenylalanine; PK, proteinase K; pOmpA, pro-OmpA; RNC, ribosome-associated nascent chain; SD, standard deviation; smORF, small open reading frame; SRP, signal recognition particle; TA, tail-anchored; TM, transmembrane domain; U-INV, urea-treated INV; YFP, yellow fluorescent protein; YohP(I4pBpa)$_{His}$, YohP containing the UV-reactive cross-linker pBpa at position 4; YohP(F27pBpa)$_{His}$, YohP containing the UV-reactive cross-linker pBpa at position 27; YohP$_{His}$, His-tagged YohP.

association allows the coordinated transfer of the RNC to the insertion site and the successive cotranslational insertion of membrane proteins into the membrane [2–4]. In contrast, secretory proteins—i.e., proteins of the periplasmic space or the outer membrane—usually follow a posttranslational targeting mode, which depends on the ATPase SecA and, in some cases, cytosolic chaperones like SecB [5, 6]. SecA associates with the SecYEG translocon and translocates these secretory proteins in ATP-dependent steps across the SecYEG channel [7, 8].

One hallmark of the cotranslational protein targeting pathway is the early recruitment of SRP to translating ribosomes [9–11]. This involves a signal sequence–independent scanning of the ribosomal exit tunnel by the C-terminal M-domain of Ffh, which is followed by the signal sequence–dependent binding of SRP to its substrates [12]. This latter step is initiated when the nascent membrane protein reaches a length of approximately 40–45 amino acids [12, 13]. At this length, the hydrophobic N-terminal signal anchor sequence is sufficiently exposed to the outside of the ribosome, while approximately 30–35 amino acids are still shielded by the ribosomal peptide tunnel. Once SRP is bound to the signal sequence, cotranslational membrane targeting of RNCs to the FtsY-SecYEG or FtsY-YidC complexes is initiated. This cotranslational targeting and insertion process avoids the accumulation of cytosolic intermediates and is particularly important for aggregation-prone membrane proteins [14].

Recent advances in genome annotation, mass spectrometry, and biochemical techniques have identified small membrane proteins as a new class of integral membrane proteins [15]. They are defined as membrane proteins of less than 50 amino acids and thus consist essentially of a single transmembrane domain (TM). They do not arise from proteolytic cleavage of larger proteins or ribosome-independent synthesis but rather are translated from small open reading frames (smORFs) that previously escaped annotation [16]. Small membrane proteins have been identified in both eu- and prokaryotes, but the exact function of most of them is unknown. They differ from membrane-active antimicrobial peptides (AMPs) that are synthesized and secreted by bacterial and eukaryotic cells [17–19], because small membrane proteins are considered to be retained in the cytoplasmic membrane of the producing cell [16, 20, 21]. Nevertheless, these small membrane proteins are surprisingly similar to AMPs, and the benefit of producing these potentially membrane-destabilizing peptides is largely unknown. In bacteria, the expression of many small membrane proteins is up-regulated when cells encounter nonfavorable conditions [22], and they therefore seem to be associated with the bacterial stress response. Upon stress conditions, the cellular concentration of some of these proteins can dramatically increase, and it was predicted that in *E. coli*, the small membrane protein YshB is present in more than 11,000 copies during stationary phase [23]. In addition, some small membrane proteins appear to act as assembly factors for larger oligomeric membrane protein complexes [24–26].

How these membrane proteins are targeted to and inserted into the cytoplasmic membrane is largely unknown. A cotranslational targeting by SRP is difficult to imagine for most small membrane proteins, because they would already be completely synthesized and released from the peptidyl transferase center of the ribosome before they were sufficiently exposed to the ribosomal surface for SRP binding [12]. Considering their strong hydrophobicity and aggregation propensity, an unassisted targeting to the membrane also appears unlikely. Alternatively, these small membrane proteins could be targeted by SecA, and indeed, SecA-dependent targeting of membrane proteins has been observed in some cases [27, 28]. The membrane insertion of some small membrane proteins has been studied in vivo using strains depleted for either SecYEG or YidC [29], the two insertion sites for membrane proteins in *E. coli* [30–32]. These studies did not reveal a single insertion mode for small membrane proteins but rather suggested a diverse requirement for SecYEG and/or YidC for insertion [29]. However, in these studies, extended tags had to be added to the client proteins for detection, increasing their

length by additional approximately 70 amino acids and thus facilitating their cotranslational targeting by SRP. In the current study, we analyzed the targeting and insertion mode of the small membrane protein YohP bearing no or only short C-terminal tags and identified a novel SRP-dependent posttranslational protein insertion pathway in bacteria.

## Results

### In vivo expression, membrane localization, and topology of small membrane proteins

The detection of small membrane proteins is hampered by their small size and strong hydrophobicity. To overcome this, a modified SDS-PAGE system was developed that allowed their detection after arabinose dependent in vivo expression by antibodies against an engineered C-terminal His-tag (Fig 1A). Of the four stress-related small membrane proteins that were analyzed, three (YohP, AzuC, and YkgR) were detectable by western blotting (Fig 1A), whereas YshB could not be detected in vivo. For YohP, two bands were observed, a weaker band migrating at approximately 4.6 kDa, which corresponds to the predicted mass of a YohP, and a stronger band migrating at approximately 9 kDa, which could reflect an SDS-resistant YohP dimer (Fig 1A). Although a membrane localization of AzuC has been predicted [29], AzuC is likely only peripherally attached to the membrane, because it contains several positively charged lysine and arginine residues and has a stretch of hydrophobic amino acids that appears to be too short to span the membrane completely (Fig 1A).

YohP was selected as the prototype for further studies, and its in vivo localization was first studied by using a C-terminally green fluorescent protein (GFP)-tagged version. Fluorescence microscopy revealed a speckled localization of YohP-GFP at the membrane, the cell poles, and the division site (Fig 1B and S1 Fig). In contrast, a yellow fluorescent protein (YFP) fusion of the integral membrane protein SecY showed largely homogeneous membrane localization (Fig 1B and S1 Fig), whereas the cytosolic protein YchF-GFP [33] was evenly distributed within the cytoplasm (Fig 1B and S2 Fig). The punctate appearance of YohP-GFP in vivo could result from aggregation or from internal membranes, which bacteria occasionally produce upon membrane protein expression [34]. However, electron microscopy did not reveal significant morphological changes or inclusion body formation when His-tagged YohP (YohP_His) was expressed, with the exception of a more condensed nucleoid (Fig 1C). The reason for the preferential YohP-GFP localization at the cell poles and the division sites is currently unknown and requires more detailed studies on the physiological function of YohP in *E. coli*.

Because the microscopic analyses did not unambiguously demonstrate a membrane localization of YohP, cell fractionation studies were performed. *E. coli* cells expressing YohP_His were lysed and separated by centrifugation into an S30 supernatant and a pellet fraction (P30), containing unbroken cells and potential inclusion bodies (Fig 1D and S3 Fig). The S30 fraction was then further separated by ultracentrifugation into an S150 supernatant containing soluble proteins and in a pellet fraction (P150) that was subsequently loaded onto a sucrose gradient and separated into the inner membrane vesicle (INV) fraction and the outer membrane fraction (OM). Immune detection revealed that the vast majority of YohP was localized in the INV fraction, similar to the membrane protein controls SecY and SecG (Fig 1D). In contrast, the cytosolic control protein YchF was exclusively found in the S150 fraction. As observed in whole cells (c.f. Fig 1A), YohP was present preferentially as a dimer in the INV fraction.

The use of epitope tags or GFP reporter fusions for validating protein expression and localization is a standard method in cell biology and biochemistry; however, since YohP's size is only 27 amino acids, adding even a small hexa-His-tag increases its size by approximately 20% and could hereby influence its membrane insertion, stability, or localization. Therefore, in

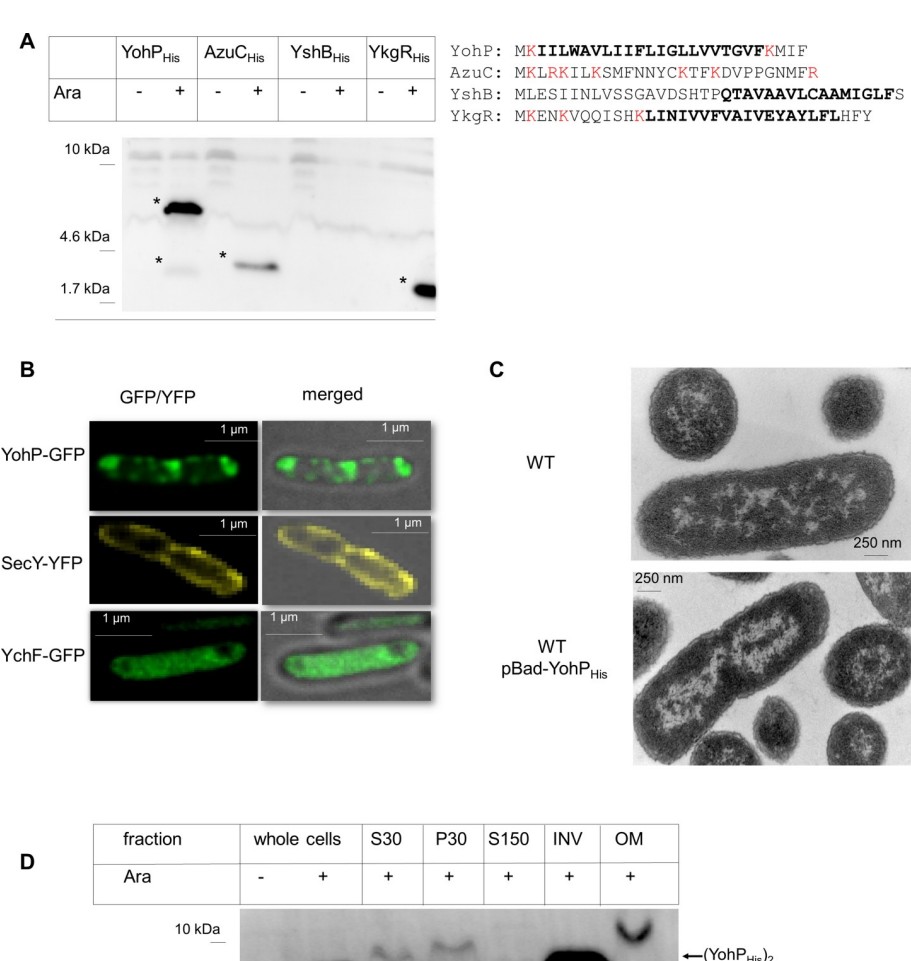

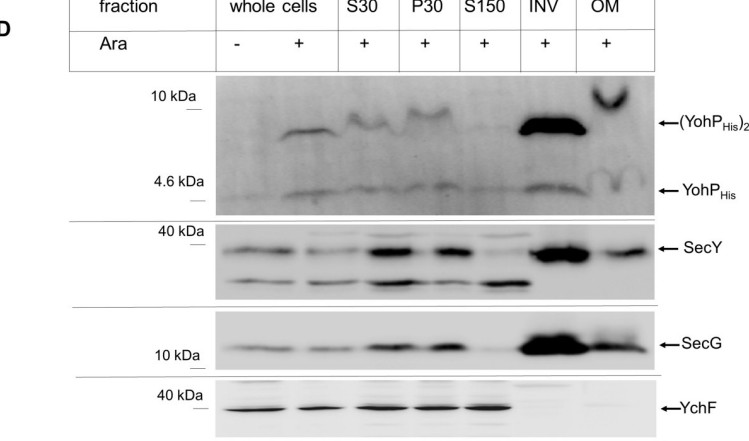

**Fig 1. Expression of small membrane proteins in E. coli and membrane localization of YohP.** (A) In vivo expression of plasmid-encoded YohP, AzuC, YshB, and YkgR. Expression was induced by 0.2% arabinose ("Ara") when indicated and whole cells were TCA precipitated, before SDS-PAGE and western blotting using α-His antibodies. The predicted amino acid sequence is shown on the right and the putative transmembrane domains are indicated in bold. Positively charged residue are shown in red. (B) YohP-GFP, SecY-YFP, and the cytosolic control protein YchF-GFP were in vivo expressed and exponentially grown cells were analyzed with a DeltaVision Ultra High Resolution Widefield Microscope (GE Healthcare, Munich, Germany) at 100× magnification. Recording, using camera sCMOS pro edge (PCO, Kelheim, Germany), was performed using a 3-μm Z-scan with 0.1-μm sectioning, and the different scans of both the fluorescence channel and the merged fluorescence/bright-field picture is shown. (C) WT *E. coli* cells and cells expressing YohP$_{His}$ from the IPTG-inducible pET19b-YohP$_{His}$ were analyzed by transmission electron microscopy. (D) *E. coli* cells expressing YohP$_{His}$ were induced with 1 mM IPTG when indicated and subsequently fractionated by differential centrifugation. S30/P30 refer to the supernatant/pellet after the first centrifugation (30,000$g$) following cell breakage. The S30 fraction was then further centrifuged (150,000$g$), and the observed pellet (P150, containing crude membranes) was subjected to sucrose gradient centrifugation, separating the INV and the OM. Of each fraction, an aliquot (40 μg protein) was separated by SDS-PAGE and, after western transfer, decorated with α-His antibodies (YohP$_{His}$), α-SecY, α-SecG, and α-YchF antibodies. A representative image of two independent biological replicates is shown. GFP, green fluorescent protein; INV, inner membrane vesicle; IPTG,

isopropyl 1-thio-β-D-galactopyranoside; OM, outer membrane fraction; TCA, trichloroacetic acid; WT, wild type; YFP, yellow fluorescent protein; YohP$_{His}$, His-tagged YohP.

vivo radioactive pulse-chase labeling was used as a tag-free detection system for YohP with or without His-tag. In vivo *T7* RNA polymerase–dependent expression of both variants was induced and cells were grown in the presence of $^{35}$S-labeled methionine/cysteine for 5 minutes, before they were chased with an excess of nonradioactive methionine/cysteine. These experiments were performed in the presence of rifampicin, which blocks *E. coli* RNA polymerase but not *T7* RNA polymerase and therefore allows the selective synthesis of YohP. For the YohP$_{His}$, once again synthesis products of 4.6 kDa and 9 kDa were detectable and stable over the 6-hour time course of the experiment (Fig 2A). Two $^{35}$S-labeled bands were also observed for the nontagged YohP but shifted to a lower molecular mass, as expected in the absence of the His$_6$-tag (Fig 2A). Like the YohP$_{His}$, the nontagged YohP was stable over the 6-hour time course. When cells carrying the empty vector were pulse-labeled in the presence of rifampicin, only one unspecific band at approximately 7 kDa was detected (Fig 2A [±], vector control), demonstrating that the rifampicin approach allows for a selective *T7* RNA polymerase–dependent expression.

When pulse-labeled cells expressing the nontagged YohP were lysed and separated into the S30, S150, and P150 fractions, YohP was preferentially found in the P150 fraction, representing the bacterial crude membrane fraction (Fig 2B). A small portion of the monomer but not of the dimer was also detectable in the soluble fraction, suggesting that YohP dimerization occurs in the membrane. Thus, both YohP$_{His}$ (Fig 1C) and nontagged YohP (Fig 2B) localize to the bacterial membrane fraction, which demonstrates that the His-tag does not significantly influence membrane localization and that it also has no significant impact on dimerization or stability of YohP.

YohP showed a strong tendency to form SDS-resistant dimers, even at low expression conditions. Dimerization of TMs is often mediated by a GxxxG motif [35, 36], which is absent in the YohP sequence. However, YohP contains the I**G**LLVVT**G**V sequence within the predicted TM (Fig 1A), which could function as an alternative dimerization motif because both glycine residues at positions 15 and 21 would be located on the same side of the helix. Single and double glycine mutants of YohP revealed that glycine residue 15 is required for YohP dimerization, whereas replacing glycine residue 21 had no significant effect (S4 Fig). This indicates that YohP dimerization does not require a canonical twin-glycine motif, but rather it is governed by a single glycine residue in the middle of the TM.

The TM of YohP is flanked by two lysine residues, which could favor a dual topology in the membrane, a feature that has been described for a few dimeric *E. coli* membrane proteins by using reporter fusions [37, 38]. The topology of YohP in the membrane was analyzed by a biochemical approach that takes advantage of the fact that the vast majority of INVs are inverted [39, 40]; i.e., the cytosolic face of the membrane is accessible to externally added proteases. Wild-type INVs and YohP$_{His}$ containing INVs were isolated, and the YohP content was analyzed by immune detection against the C-terminal His-tag (Fig 2C, lanes 1 and 2). As a control, the samples were also probed with polyclonal antibodies against the single-spanning membrane protein YfgM [41]. When INVs were treated with proteinase K (PK), only two weak bands were detectable by α-His antibodies (Fig 2C, lane 3), indicating that the His-tag of YohP was largely cleaved off by PK. For YfgM, a faster migrating species was detectable (Fig 2C, lane 3), in line with cleavage of the cytosolically exposed N-terminal 20 amino acids. We repeatedly observed only about 90% cleavage for YohP$_{His}$, which could indicate that a small fraction of YohP is oriented with the C-terminus in the periplasm. Alternatively, the cytosolically exposed

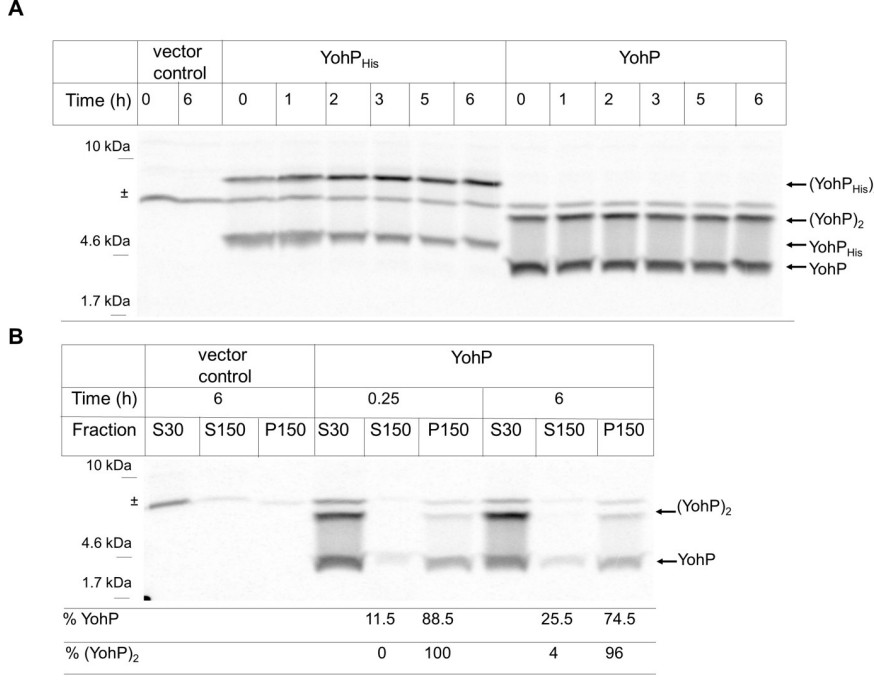

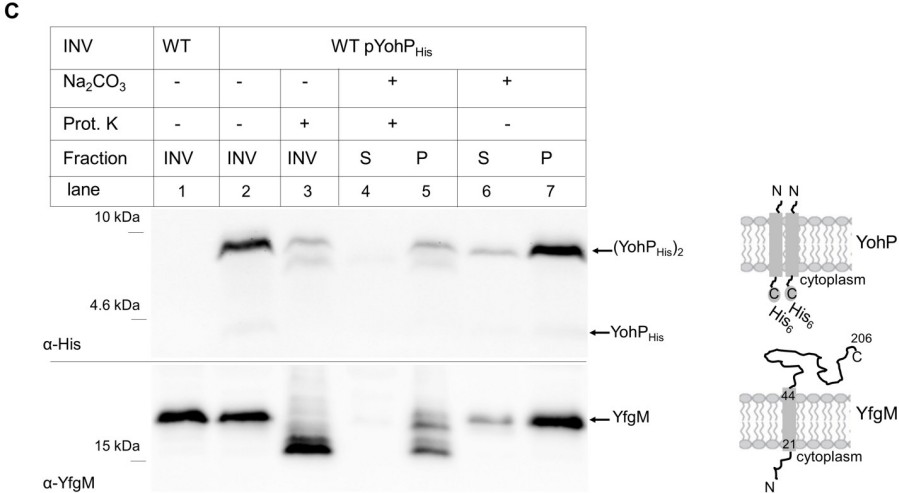

**Fig 2. Membrane localization of YohP is not influenced by the His-Tag.** (A) YohP and YohP$_{His}$ were in vivo expressed and pulse-labeled with $^{35}$S methionine/cysteine for 5 minutes. Whole cells were then TCA precipitated after the indicated time points, separated by SDS-PAGE, and YohP$_{His}$/YohP was detected by autoradiography. Indicated are the monomeric and dimeric versions. $^{\pm}$ refers to a nonspecifically labeled band that was also detected in the control sample, containing cells without plasmid. (B) YohP was in vivo expressed and pulse-labeled as above. Cells were disrupted by ultrasonic treatment and subjected to cell fractionation into an S30, S150, and P150 extract, as described in the legend to Fig 1. Quantification was performed on three independent experiments, and the relative mean values of monomeric and dimeric YohP found in the supernatant (S150) or the membrane fraction (P150), respectively, are shown. Underlying data for this figure can be found in S1 Data. (C) INVs of WT cells and cells expressing YohP$_{His}$ were treated directly with proteinase ("Prot.") K or only after carbonate extraction and separation into carbonate-resistant integral membrane proteins ("P") and carbonate-sensitive membrane-associated proteins ("S"). In addition, carbonate extraction was performed without subsequent proteinase K treatment. YohP was detected by its C-terminal His-tag, whereas the control membrane protein YfgM was detected by polyclonal antibodies raised against the native protein. A representative gel of at least two independent biological replicates with at least two technical replicates is shown. INV, inner membrane vesicle; TCA, trichloroacetic acid; WT, wild type; YohP$_{His}$, His-tagged YohP.

His-tag could be partially shielded, e.g., by contacting the polar head groups of phospholipids. This was further analyzed by alkaline carbonate extraction, which disintegrates the INVs into linear membrane sheets and thus makes both sides of the membrane accessible to PK. INVs were first treated with carbonate and separated by centrifugation into carbonate-resistant integral membrane proteins (Fig 2C, lane 5 ["P"]) and carbonate-sensitive membrane-associated proteins (Fig 2C, lane 4 ["S"]). Both fractions were then subjected to PK treatment. α-His antibodies detected again the two weak bands for the carbonate-resistant YohP (Fig 2C, lane 5), suggesting that the His-tag is indeed partially shielded even in membrane sheets. No signal was detected in the carbonate-sensitive fraction (Fig 2C, lane 4). For YfgM, a significant but also not complete proteolysis was observed in membrane sheets (Fig 2C, lane 5). Analyzing these membranes by just carbonate extraction clearly demonstrated that YohP is an integral membrane protein because more than 90% were found to be carbonate resistant like the integral membrane protein YfgM (Fig 2C, lane 7). In summary, these data show that the vast majority of YohP$_{His}$ is oriented in a C$_{in}$-N$_{out}$ topology, although we cannot entirely exclude that a very small portion is oriented in N$_{in}$-C$_{out}$ topology.

## YohP insertion into the bacterial membrane depends on the SRP pathway in vitro

The majority of bacterial membrane proteins are inserted by either the SecYEG translocon or the YidC insertase after they have been targeted to the membrane by the SRP pathway [31]. However, the insertion requirements for small membrane proteins are largely unknown [29]. For studying this, a purified membrane-free in vitro transcription/translation system (cytosolic translation factor [CTF] system [39]) in combination with inside-out INVs or liposomes [39] was used. This system allows one to determine the membrane insertion mechanism of in vitro–synthesized and radioactively labeled YohP and YohP$_{His}$ by analyzing their PK protection.

When YohP was in vitro synthesized in the absence of INV and treated with PK, more than 90% of the in vitro–synthesized YohP was degraded (Fig 3A). The small fraction of YohP that was still detectable after PK treatment probably reflects some protease-resistant aggregates. When PK treatment was performed on YohP that was synthesized in the presence of INV, more than 80% of YohP was PK-resistant, indicating that YohP was inserted into INVs in vitro and thus not accessible to PK cleavage. YohP synthesis in the presence of liposomes, composed of *E. coli* phospholipids at the native phospholipid composition (70% phosphatidylethanolamine, 25% phosphatidylglycerol, and 5% cardiolipin), did not confer strong PK resistance. This indicates that YohP has only a weak tendency to spontaneously insert into liposomes. The same experiment was also repeated with YohP$_{His}$, which showed a strong PK protection in the presence of INV but only a weak PK protection in the absence of INV or in the presence of liposomes (Fig 3A). The PK-protected fragment of YohP$_{His}$ migrated slightly faster on SDS-PAGE than YohP$_{His}$ and showed the same migration pattern as the PK-treated nontagged YohP. Thus, PK cleaves off the His-tag from YohP$_{His}$, which indicates that the C-terminus of YohP is facing the cytosolic face of the inverted membrane vesicles and is accessible to PK treatment. This supports the C$_{in}$-N$_{out}$ orientation that was predicted based on the carbonate extraction data (Fig 2). In summary, membrane insertion of YohP/YohP$_{His}$ does not efficiently occur spontaneously in our assays, suggesting that it requires a dedicated targeting or insertion machinery, or both, that is provided by the INVs.

This was further analyzed by in vitro insertion assays using urea-treated INVs (U-INVs). Urea treatment removes a significant portion of membrane-associated proteins, including the known membrane-associated targeting factors FtsY, SRP, and SecA (S5 Fig), but has only a

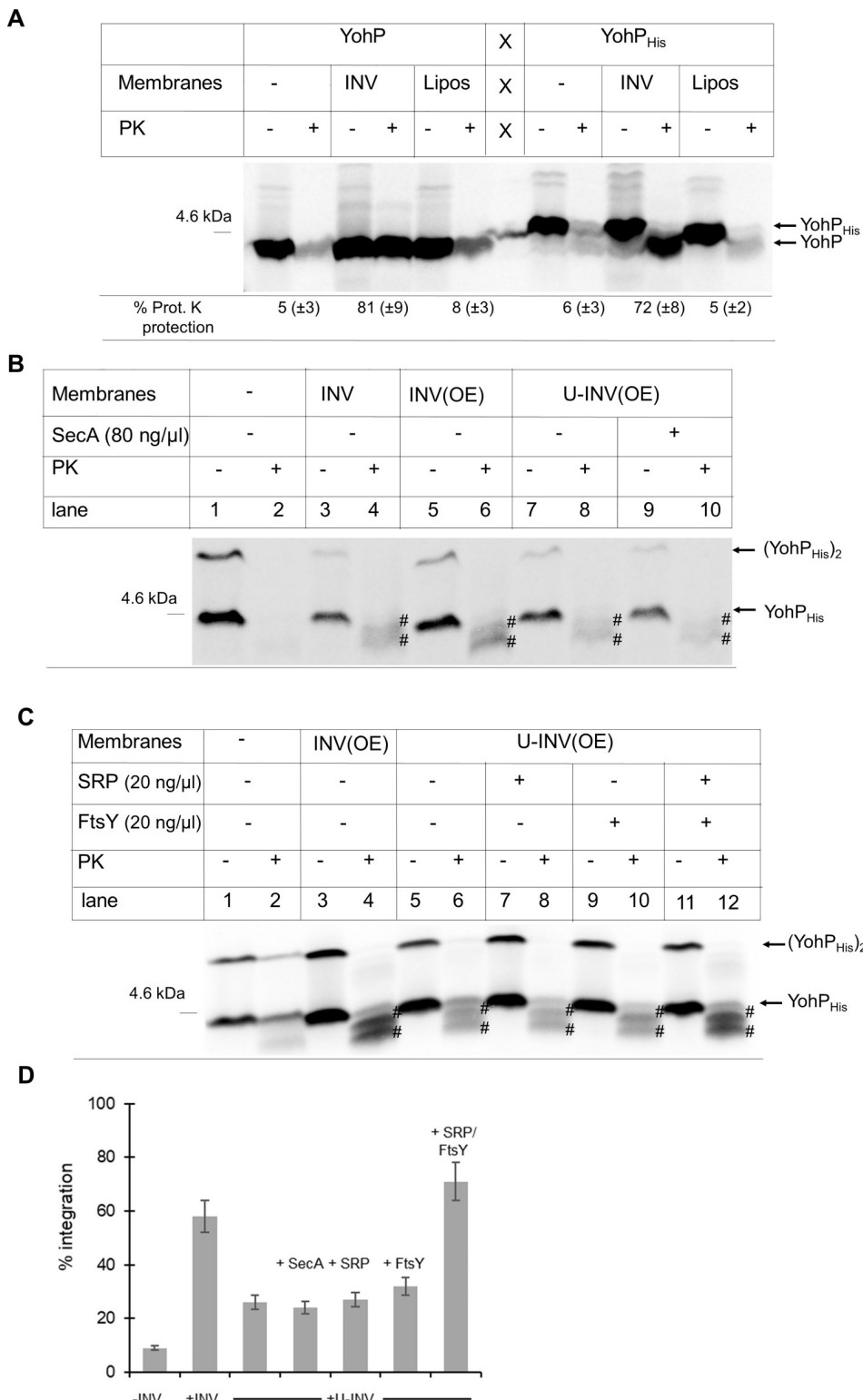

**Fig 3. YohP does not spontaneously insert into the membrane but requires the SRP pathway.** (A) YohP and YohP$_{His}$ were in vitro synthesized using a purified coupled transcription/translation system (CTF system). In vitro synthesis was performed in the presence of inverted INVs or liposomes ("Lipos") or buffer, as a control. Liposomes were generated from *E. coli* phospholipids and contained 70% PE, 25% PG, and 5% CL. After 20 minutes of in vitro synthesis, half of the sample was directly TCA precipitated, whereas the other half was first treated with PK before TCA

precipitation. Samples were then separated by SDS-PAGE and analyzed by autoradiography. Indicated are the in vitro–synthesized YohP and YohP$_{His}$. The quantification is based on three independent experiments and the mean values (±SD) are shown. Underlying data for this figure can be found in S1 Data. (B) YohP$_{His}$ was in vitro synthesized as above, in the presence of either wild-type INV, INV(OE), or U-INV(OE). When indicated, purified SecA (80 ng/μl final concentration) was added. The PK-protected fragments of YohP$_{His}$ are indicated (#). Please note that in these experiments, the amount of in vitro–synthesized YohP was reduced for preventing any saturation effects. (C) As in (B), but purified SRP, FtsY, or both (20 ng/μl each final concentration) were added when indicated. (D) Quantification of YohP insertion using the different conditions described in (B) and (C). The values are means of three independent experiments and the SD is indicated by error bars. Underlying data for this figure can be found in S1 Data. CL, cardiolipin; CTF, cytosolic translation factor; INV, inner membrane vesicle; INV(OE), INV from a SecYEG-overproducing *E. coli* strain; PE, phosphatidylethanolamine; PG, phosphatidylglycerol; PK, proteinase K; SD, standard deviation; SRP, signal recognition particle; TCA, trichloroacetic acid; U-INV, urea-treated INV; YohP$_{His}$, His-tagged YohP.

moderate effect on the activity of the SecYEG/YidC insertion machinery [39]. As the CTF-type in vitro translation system is also largely devoid of SRP/FtsY and SecA, protein transport into U-INVs is greatly enhanced by the addition of purified targeting factors. YohP$_{His}$ was in vitro synthesized in the presence of wild-type INV and INV derived from a strain overexpressing SecYEG (INV[OE]). In these experiments, the amount of in vitro–synthesized YohP was reduced in order to prevent possible saturation effects that might obscure any membrane insertion defect. In the presence of INV, YohP$_{His}$ showed PK protection as above, but the lower signal allowed for the detection of two PK-protected fragments that migrated slightly faster than YohP$_{His}$. This indicates that PK can cleave the C-terminal His-tag at two distinct positions. The INV(OE) showed slightly improved YohP insertion compared to wild-type INV (Fig 3B, compare lane 4 with lane 6), pointing to an involvement of the SecYEG translocon in YohP insertion. When INV(OE) were urea-treated, YohP insertion was significantly reduced (Fig 3B, lane 8) and was not restored by the addition of purified SecA (Fig 3B, lane 10). As a control, the translocation of the SecA-dependent protein OmpA into U-INV was analyzed and showed strongly enhanced translocation into U-INV when purified SecA was added (S5 Fig). These data indicate that SecA is not required for targeting or insertion of YohP into the bacterial membrane.

Next, purified SRP and FtsY were added to U-INV, and YohP insertion was analyzed. Adding either SRP or FtsY alone did not improve insertion of YohP into U-INV (Fig 3C, lanes 8 and 10); however, when both SRP and FtsY were added, YohP insertion was comparable to the insertion into nontreated INV (Fig 3C, lane 12). As a control, the insertion of the SRP-dependent model membrane protein mannitol permease (MtlA) was analyzed, which showed increased insertion upon the addition of SRP/FtsY (S5 Fig). Quantification of several independent experiments clearly demonstrated that only the combined addition of both SRP and its receptor FtsY was able to stimulate YohP insertion into U-INV (Fig 3D). This suggests that YohP insertion does not merely depend on a putative chaperone function of SRP but rather requires the complete targeting function of the bacterial SRP system.

## Posttranslational membrane insertion of YohP by the SRP

A key hallmark of SRP-dependent membrane protein insertion is that ribosome-bound SRP recognizes its substrates while they are synthesized [14]. However, YohP with just 27 amino acids appears to be too short for a cotranslational recognition by SRP and is likely released from the peptidyl transferase center of the ribosome before SRP can bind to it. The SRP/FtsY-dependent insertion of YohP therefore rather suggests a posttranslational function of SRP/FtsY during YohP insertion. This was confirmed by uncoupling protein synthesis from membrane insertion. YohP was synthesized in vitro in the absence of membranes, and translation was then stopped by the addition of chloramphenicol, followed by centrifugation to remove

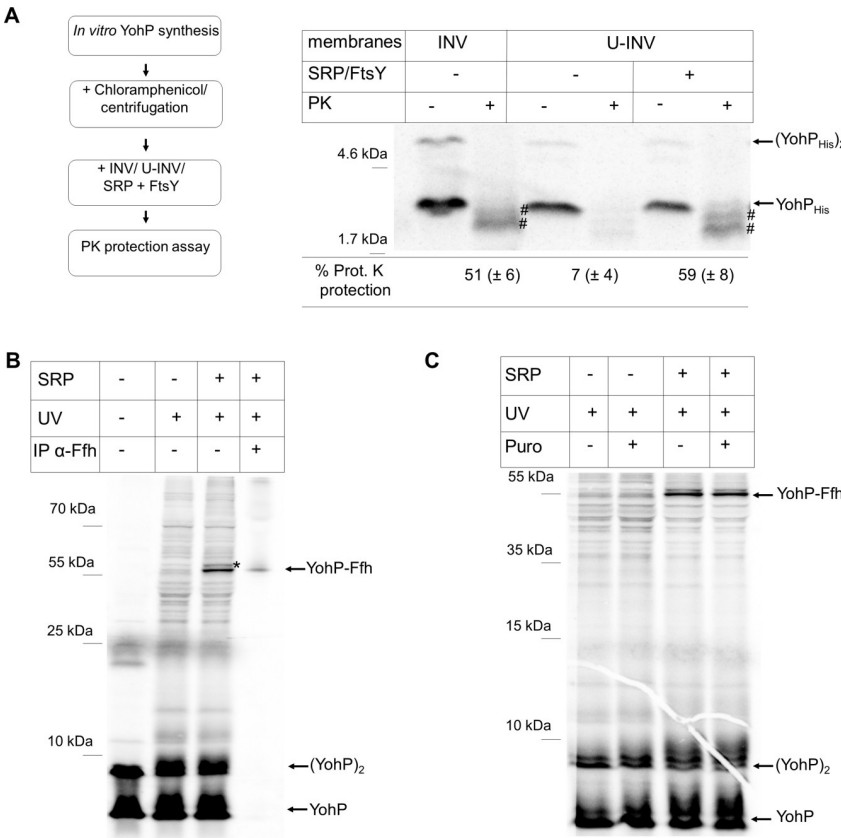

**Fig 4. Posttranslational recognition and insertion of YohP by SRP.** (A) YohP was in vitro synthesized in the absence of INV and translation was terminated by the addition of chloramphenicol (35 mg/ml). Samples were centrifuged for removing ribosomes and aggregates, and the supernatant was incubated with INV or U-INV for 10 minutes. When indicated, purified SRP and FtsY were added together with the U-INV. Samples were then processed as described in the legend to Fig 3. Quantification is based on three independent experiment, and the SD is shown in brackets. Underlying data for this figure can be found in S1 Data. (B) YohP(I4pBpa)$_{His}$ was in vitro synthesized and radioactively labeled. When indicated, YohP(I4pBpa)$_{His}$ was incubated with purified SRP (100 ng/µl) and UV-exposed. Samples were then TCA precipitated and separated by SDS-PAGE and analyzed by autoradiography. When indicated, a 10-fold scaled-up reaction was subjected to cross-linking and subsequently immune-precipitated by α-Ffh antibodies, covalently coupled to sepharose beads. (C) Cross-linking was performed as in (B), but when indicated, puromycin (1 mM) ("Puro") was added prior to UV exposure for dissociating the ribosome. INV, inner membrane vesicle; IP, immune precipitation; PK, proteinase K; SD, standard deviation; SRP, signal recognition particle; TCA, trichloroacetic acid; U-INV, urea-treated INV; YohP(I4pBpa)$_{His}$, YohP containing the UV-reactive cross-linker para-benzoyl-L-phenylalanine at position 4.

ribosomes and any potential protein aggregates (Fig 4A). The supernatant after centrifugation was then incubated with INVs or U-INVs in the absence or presence of SRP/FtsY. In the presence of INV, efficient membrane insertion of YohP was observed, but only weak insertion into U-INV was seen (Fig 4A). Adding SRP/FtsY together with the U-INV strongly stimulated YohP$_{His}$ insertion (Fig 4A), as judged by PK resistance. This shows that YohP is a unique SRP substrate, because SRP executes its role posttranslationally.

Although the data described above clearly indicate that YohP insertion is SRP-dependent, proof for a direct interaction between YohP and SRP was still missing. This was addressed by using an in vitro site-directed cross-linking approach. A TAG stop codon was inserted at position 4 of the *yohP* coding sequence, and the corresponding plasmid was used in an S-135 in vitro translation system derived from an *E. coli* strain expressing a TAG amber suppressor tRNA and a cognate tRNA synthetase. This allows the in vitro charging of the suppressor

tRNA with the UV-sensitive cross-linker para-benzoyl-L-phenylalanine (pBpa) [42]. During in vitro synthesis in the presence of pBpa, pBpa was then inserted at position 4 of the radioactively labeled YohP (YohP[I4pBpa]). In contrast to the CTF in vitro system, the S-135 system contains all soluble *E. coli* proteins but is largely devoid of membranes. In vitro–synthesized YohP(I4pBpa) was incubated with purified, nonradioactive SRP and UV-exposed for inducing the cross-link reaction (Fig 4B). Phosphor imaging revealed a strong UV-dependent band at approximately 55 kDa, potentially reflecting a cross-link between Ffh, the 48 kDa protein component of SRP, and radiolabeled YohP (Fig 4B). This was confirmed by immune precipitation with α-Ffh antibodies of an in vitro cross-linking reaction that had been scaled up 10-fold, which specifically precipitated the 55-kDa band (Fig 4B), as visualized by phosphor imaging.

If SRP binding to YohP was indeed a posttranslational event, then it should occur after YohP is released from the ribosome. This was tested by inducing the cross-linking of in vitro–synthesized YohP(I4pBpa) after the addition of puromycin, a tRNA mimic that dissociates actively translating ribosomes [43]. The formation of the SRP-YohP cross-linking product was not influenced by the addition of puromycin (Fig 4C), indicating that the SRP-YohP contact is ribosome-independent. As a control, the effect of puromycin on SRP cross-links to RNCs of the typical cotranslational SRP substrate MtlA was analyzed, and the data showed that cross-links between SRP and MtlA nascent chains disappeared upon puromycin treatment (S6 Fig).

Cotranslational contact of SRP to its substrates is generally only observed when substrates are longer than approximately 40–45 amino acids [10, 12, 13, 44], explaining why YohP is not cotranslationally recognized by SRP. On the other hand, SRP can bind to translating ribosomes even before the critical length of 40–45 amino acids is reached [9, 12]. In order to further test whether small membrane proteins are only recognized posttranslationally by SRP, the in vitro cross-link pattern of YohP(I4pBpa) was compared with the cross-link pattern of YohP (F27pBpa), which contained pBpa at its very C-terminus. In both cases, robust adducts of 55 kDa were detected (Fig 5A, compare lanes 3, 6, 9, and 11). Importantly, cross-links to SRP from residue 27 can only occur after YohP had been released from the ribosome.

For YohP(I4pBpa), the 55-kDa band was visible upon UV exposure in the presence of SRP (Fig 5A, compare lanes 1, 2, and 3), but this SRP-YohP cross-link was strongly reduced in the presence of INV (Fig 5A, lane 4). This indicates that the YohP-SRP contact is lost when YohP is inserted into the membrane, as previously established for typical SRP substrates [45]. As shown before (Fig 4C), the addition of puromycin before UV exposure did not prevent the SRP-YohP cross-link (Fig 5A, lane 6). Adding puromycin at the beginning of the in vitro reaction, on the other hand, blocked protein synthesis completely, demonstrating that puromycin was active in dissociating the ribosome (Fig 5A, lane 5). The same conditions were then tested for YohP(F27pBpa). Upon UV exposure in the presence of SRP, the 55 kDa was visible (Fig 5C, lane 9) and disappeared in the presence of INV (Fig 5C, lane 10) but not in the presence of puromycin (Fig 5C, lane 11). Taken together, these data demonstrate that SRP recognizes YohP after it is released from the ribosome, i.e., posttranslationally.

In order to establish whether SRP is involved in the insertion of other small membrane proteins, the in vitro cross-linking experiment was also performed with YkgR (Fig 1A). YkgR containing pBpa at position 6 was in vitro synthesized, and cross-linking was induced as described above. As seen for YohP, YkgR(V6pBpa) showed a UV- and SRP-dependent cross-link product at approximately 55 kDa, which was not influenced by the addition of puromycin but disappeared in the presence on INV (S6 Fig). Thus, posttranslational recognition by SRP is likely a general attribute of small membrane proteins.

In order to address the possibility that the detection of cross-linking products we observe in the presence of purified SRP might result from high/nonphysiological SRP concentrations, cross-linking was also repeated in vivo using wild-type *E. coli* cells with native SRP

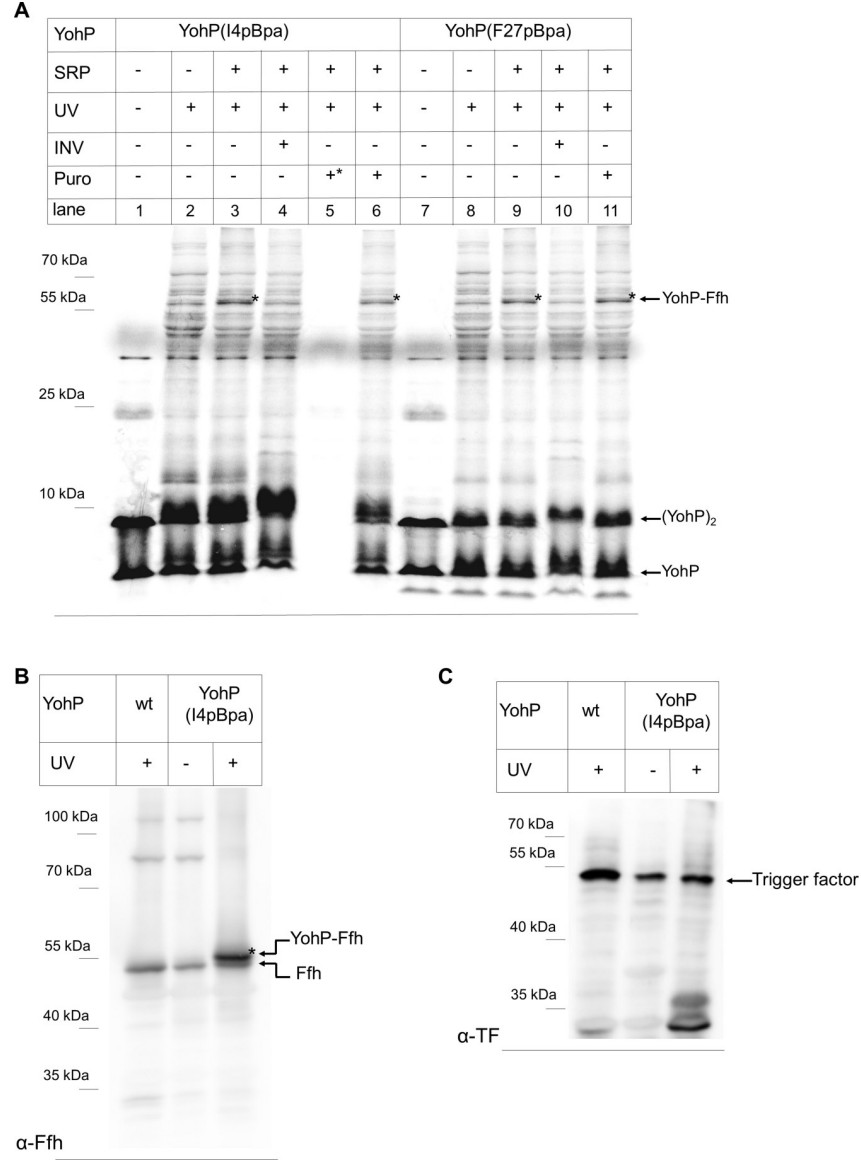

**Fig 5. SRP binds to YohP in vitro and in vivo.** (A) Either YohP(I4pBpa)$_{His}$ or YohP(F27pBpa)$_{His}$ were in vitro synthesized in the presence or absence of purified SRP (100 ng/μl) and UV-exposed. When indicated, UV exposure was performed in the presence of INV or after puromycin treatment ("Puro"). Lane 5 contains a sample in which puromycin was added at the start of the in vitro synthesis (+*) for controlling its ribosome dissociating activity. (B) YohP$_{His}$ and YohP(I4pBpa)$_{His}$ were in vivo expressed in wt *E. coli* cells and when indicated, in vivo cross-linking in whole cells was induced by UV exposure. YohP and its cross-linked partner proteins were subsequently enriched by a single-step metal-affinity purification, TCA precipitated, and analyzed by western blotting using α-Ffh antibodies. Indicated are the copurifying Ffh and the Ffh-YohP cross-linking product. (C) The same material as in (B) was decorated with antibodies against the cytosolic chaperone Trigger factor. Indicated is the copurifying Trigger factor, but no specific cross-linking product was observed. INV, inner membrane vesicle; SRP, signal recognition particle; TCA, trichloroacetic acid; wt, wild type; YohP(I4pBpa)$_{His}$, YohP containing the UV-reactive cross-linker para-benzoyl-L-phenylalanine at position 4; YohP(F27pBpa)$_{His}$, YohP containing the UV-reactive cross-linker para-benzoyl-L-phenylalanine at position 27; YohP$_{His}$, His-tagged YohP.

concentrations. YohP(I4pBpa) was expressed in vivo, and whole cells were irradiated with UV light. Subsequently, a cell extract was prepared, and YohP and its cross-linked partner proteins were enriched by a single-step metal-affinity purification via the C-terminal His-tag on YohP.

The material was then subjected to western blotting followed by immune detection using α-Ffh antibodies. This revealed the presence of the 55-kDa Ffh-YohP cross-linking product migrating just above the endogenous Ffh, which was also present in the only partially purified material (Fig 5B). Thus, YohP is bound by SRP not only in vitro but also in living *E. coli* cells. To further validate the specificity of the SRP-YohP cross-link detected, the same material was also decorated with antibodies against the cytosolic chaperone Trigger factor, a highly abundant peptidyl-prolyl isomerase [46–48]. However, we were unable to obtain any evidence of cross-linking products formed between YohP and Trigger factor (Fig 5C), despite the fact that Trigger factor is about 50 times more abundant than SRP [47] and was easily detectable in the partially purified material. These data further corroborate our hypothesis that the small membrane protein YohP is a specific SRP substrate both in vitro and in vivo.

## YohP is inserted into the bacterial membrane by the SecYEG translocon or YidC

SRP targets bacterial membrane proteins to either the SecYEG translocon or to the YidC insertase [31], but whether this also applies to the posttranslational targeting of small membrane proteins is unknown. This issue was first analyzed by using the glycoresin Ipomoeassin F (IpomF). IpomF has been shown to inhibit the protein transport activity of the eukaryotic Sec61 complex [49], but whether it also inhibits the homologous SecYEG complex has not been studied so far. The effect of IpomF was first tested on the strictly SecYEG-dependent secretory protein OmpA. The addition of increasing IpomF concentration to the in vitro transport assay resulted in a concentration-dependent decrease of PK protection (Fig 6A). OmpA contains a cleavable signal sequence, which is not completely removed in vitro because of the reduced activity of signal peptidase in INV [50]. Therefore, two PK-protected fragments were observed (Fig 6A), corresponding to mature OmpA (OmpA) and pro-OmpA (pOmpA). Intriguingly, although IpomF almost completely blocked translocation of OmpA, there was no major difference in signal sequence cleavage in the absence or presence of IpomF. This is in line with previous data, showing that processing can occur prior to complete translocation [50]. As the catalytic site of signal peptidase is located on the periplasmic side of the membrane, unimpaired signal sequence cleavage indicates that IpomF does not inhibit the initial contact between OmpA and the SecYEG translocon but rather the later steps of OmpA translocation.

The insertion of YohP was also inhibited by IpomF (Fig 6B and S7 Fig), albeit that the inhibitory effect of IpomF on YohP insertion was less pronounced than on OmpA translocation. Unfortunately, higher IpomF concentrations could not be tested in the in vitro system, because of the propensity of IpomF to precipitate. The effect of IpomF on the insertion of the multispanning membrane protein MtlA was also tested, but here we did not observe a significant inhibition (Fig 6B and S7 Fig). As MtlA can be inserted by either the SecYEG translocon or the YidC insertase [51], MtlA insertion would still be possible if IpomF inhibits only the SecYEG translocon. Thus, whereas our data demonstrate that the natural cytotoxin IpomF can be used to inhibit transport of strictly SecYEG-dependent substrates in *E. coli*, this requires significantly higher concentrations than in the eukaryotic Sec61 system, and interpretation of the data is complicated by the presence of two insertion sites for membrane proteins in the bacterial membrane. It should also be noted that we can currently not exclude that IpomF inhibits not only the SecYEG translocon but also SecA, which would explain why the inhibitory effect is more pronounced for the SecA-dependent substrate OmpA.

To provide further information about the insertion mode of YohP, the trimeric SecYEG translocon and the YidC insertase were purified and reconstituted individually into

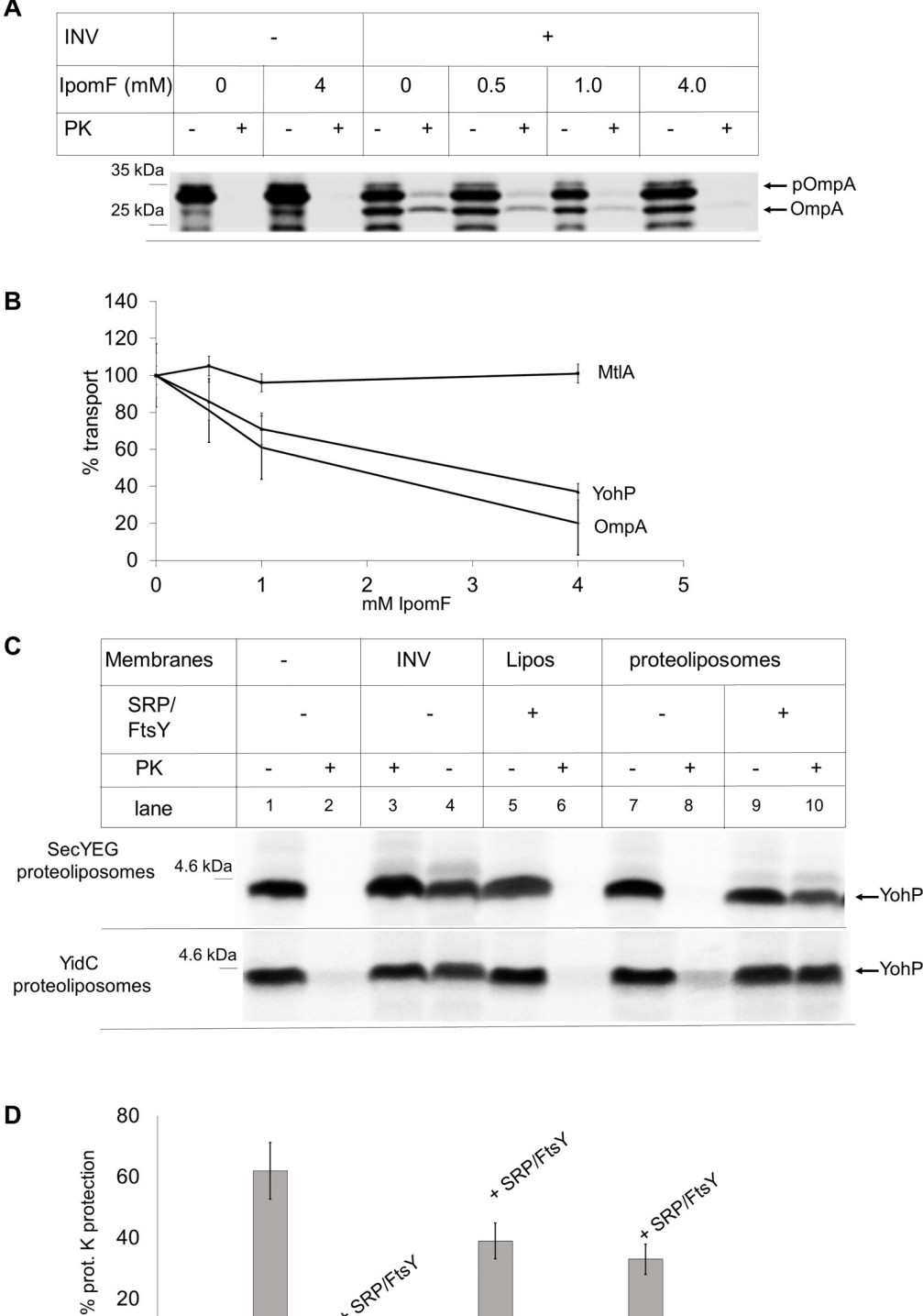

**Fig 6. SRP-dependent insertion of YohP via the SecYEG translocon or YidC.** (A) The SecYEG-dependent secretory protein OmpA was in vitro synthesized in the absence or presence of INV that were pretreated with the inhibitor IpomF or with DMSO as control (0 mM IpomF). Samples were then subjected to PK treatment. Indicated are pOmpA and mature OmpA, which occurs after signal sequence cleavage. (B) Quantification of OmpA, MtlA, and YohP transport in IpomF-treated INV. Transport in the absence of IpomF was set to 100%. Indicated are the mean values of

at least three (MtlA, OmpA) or five (YohP) independent experiments, and the SD is shown by error bars. Underlying data for this figure can be found in S1 Data. (C) YohP was in vitro synthesized in the absence of membranes, translation was stopped by the addition of chloramphenicol, and the sample was centrifuged for removing aggregates. The supernatant was subsequently incubated with INV, liposomes ("Lipos"), or proteoliposomes containing the SecYEG complex (100 ng/μl) or the YidC insertase (100 ng/μl) and in the presence of absence of purified SRP/FtsY (20 ng/μl, each). Samples were then subjected to PK treatment as before. (D) Quantification of at least three independent experiments performed as in (C). The SD is indicated by error bars. INV, inner membrane vesicle; IpomF, Ipomoeassin F; MtlA, mannitol permease; PK, proteinase K; pOmpA, pro-OmpA; SD, standard deviation; SRP, signal recognition particle.

proteoliposomes containing the natural *E. coli* phospholipid composition as above (c.f. Fig 3). These proteoliposomes were then used to analyze the insertion of in vitro–synthesized YohP, after synthesis had been terminated by chloramphenicol and centrifugation as described above (Fig 4). In these experiments, we employed the nontagged YohP variant in order to exclude any potential effect of the His-tag on the membrane insertion mode. PK protection of YohP was observed in INV but not in the absence of INV (Fig 6C, lanes 2 and 4). YohP was also not PK protected in the presence of liposomes, even if SRP and FtsY had been added (Fig 6C, lane 6 and S8 Fig). Furthermore, the reconstituted SecYEG proteoliposomes were unable to insert YohP, unless purified SRP/FtsY was also added (Fig 6C, compare lanes 8 and 10). Adding just SRP or FtsY did not lead to PK protection of YohP and neither did the addition of purified Hsp70 or SecA to SecYEG proteoliposomes (S8 Fig). These findings are in agreement with the SRP/FtsY-dependent and SecA-independent insertion of the YohP$_{His}$ into U-INV (Fig 3). Interestingly, the SRP/FtsY-dependent insertion of YohP was also observed with the reconstituted YidC proteoliposomes (Fig 6C, lower panel and Fig 6D). Thus, our in vitro data suggest that SRP can target YohP to either the SecYEG translocon or the YidC insertase for insertion, as previously observed for other membrane proteins lacking large periplasmic loops [51].

## YohP targeting is likely initiated independent of translation

In combination, our data demonstrate that YohP engages the SRP pathway for its posttranslational membrane insertion. This finding raises the question of whether the SRP pathway is involved in targeting the newly synthesized YohP from the cytosol to the membrane, or whether SRP/FtsY is primarily required for the insertion of already membrane-localized YohP. The latter would argue for the existence of an alternative membrane targeting strategy for small membrane proteins.

Translation-independent membrane targeting has recently been observed for some bacterial mRNAs encoding membrane proteins [52, 53], although how these mRNAs are subsequently handled by the ribosome and how the resulting translation product is inserted into the membrane are unknown. On this basis, the possibility of an mRNA targeting step during YohP membrane insertion was analyzed by monitoring *yohP* mRNA localization in vivo using the established MS2 reporter system [54]. In this system, a target mRNA lacking the ribosome binding site but containing a hexa-repeat stem-loop sequence at its 3′-end is coexpressed with a fluorescently labeled MS2 phage protein. The MS2 protein binds with high specificity to its cognate stem-loop sequence and the MS2-mRNA complex can then be monitored by fluorescence microscopy.

When MS2-GFP was expressed in *E. coli* in the absence of a target mRNA, it localized almost exclusively to the bacterial cytoplasm and showed some clustering, which probably reflects the formation of aggregates (Fig 7A). The coexpression of MS2-GFP together with the mRNA of the cytosolic protein BglB resulted in an almost homogeneous distribution of the fluorescence signal within the cytosol (Fig 7B). This was confirmed by membrane staining

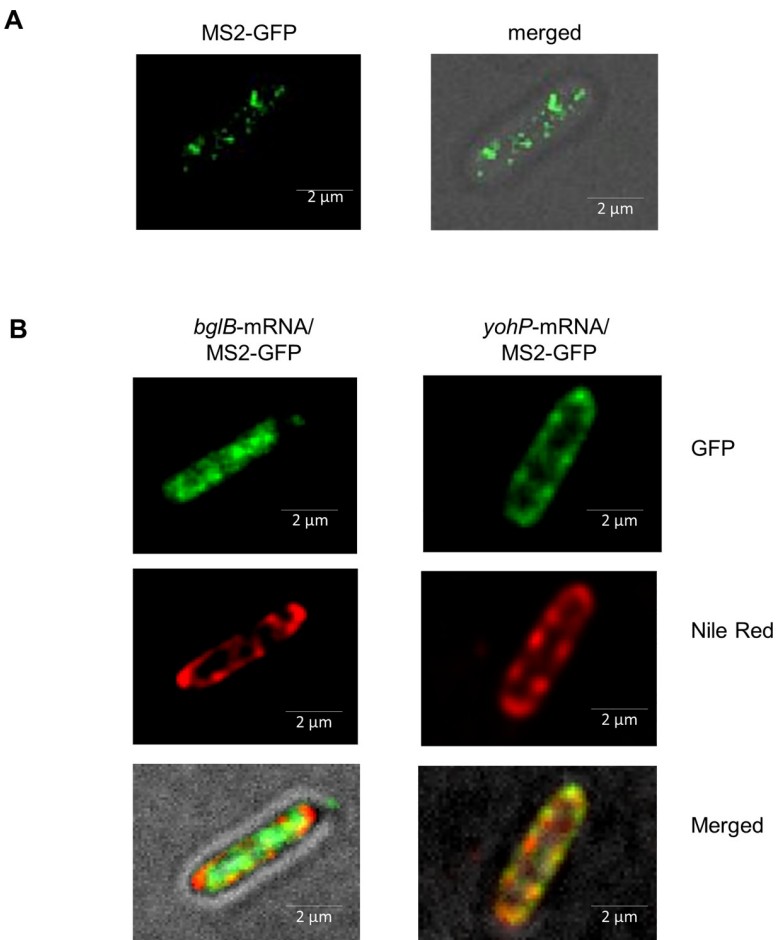

**Fig 7. Translation-independent membrane enrichment of the YohP mRNA. (A)** *E. coli* wild-type cells carrying pBad24-MS2-GFP were induced with 0.1 mM arabinose and 5 μL culture solution were placed on a sterile glass-bottom dish (35-mm dish with 20-mm bottom well, glass thickness 0.16–0.19 mm). Imaging was performed with the DeltaVision Ultra microscope (exposure 0.2 seconds for GFP at 35% laser intensity and 0.075 seconds for bright field at 5% laser intensity), and 3-μm Z-scans were recorded with an interval of 1 μm. The displayed images were taken at the focal point. The image was developed with the ImageJ Fiji software. (B) As in (A) but cells were coexpressing the pSC-bglB-MS2.6x plasmid or the pSC-yohP-MS2.6x plasmid, encoding the *bglB* or *yohP* mRNA, respectively, each with a deleted ribosome binding site and a hexa-repeat MS2 stem-loop recognition sequence at the 3′ UTR. Expression of the respective mRNAs was induced at an $OD_{600}$ = 0.5 with 4 mM IPTG, and MS2-GFP was induced 40 minutes after IPTG addition with 0.1 mM arabinose. Samples were analyzed as above and Nile red was used to stain the membrane. Nile red staining was monitored with an exposure time of 0.2 seconds at a laser intensity of 35%. GFP, green fluorescent protein; IPTG, isopropyl 1-thio-β-D-galactopyranoside.

with the lipophilic fluorescent dye Nile red, which allows the membrane boundaries of the bacterial cell to be localized. Merging the GFP signal and the Nile red signal showed virtually no overlap, demonstrating that the MS2-GFP-*bglB* mRNA complex is preferentially localized in the cytosol. In contrast, when the *yohP* mRNA was coexpressed with MS2-GFP, the majority of the GFP signal—i.e., the MS2-GFP-*yohP*-mRNA complex—was detected at the membrane and colocalized with the Nile red signal (Fig 7B). These data are in line with the reported existence of an mRNA targeting pathway operating in bacterial cells [52, 53].

Our combined data raise the intriguing possibility not only that SRP targets YohP posttranslationally to the membrane but also that it facilitates the posttranslational membrane insertion of YohP after the *yohP* mRNA is targeted to the membrane and translated. Such a mechanism

would define a so far unknown strategy for handling unconventional membrane proteins and merits further exploration.

## Discussion

In our current study, we have identified an unexpected SRP-dependent posttranslational insertion mode for the small membrane protein YohP. Bacterial small membrane proteins represent a novel and largely unexplored class of membrane proteins, and how they are handled by the bacterial protein targeting machinery has not been studied so far, to our knowledge.

Owing to improved analyses and annotation tools, the number of small proteins in pro- and eukaryotes is rapidly increasing. This is exemplified by analyses of the mouse proteome, which identified 3,701 proteins of less than 100 amino acids [55, 56]. A survey among 532 sequenced genomes of prokaryotes identified approximately 27,000 annotated protein sequences of less than 50 amino acids [57]. Among them are many hydrophobic proteins, which are just long enough to span a membrane bilayer [29, 58]. One example is YohP, which was studied here and which is highly enriched in the inner membrane of *E. coli* (Fig 1). YohP shows a carbonate-resistant interaction with the *E. coli* membrane (Fig 2), which defines it as an integral membrane protein. Bacterial small membrane proteins constitute an interesting class of membrane proteins, because they are often enriched in the membrane upon stress conditions, but are also highly similar to AMPs, which induce membrane damage by pore formation or phase separation [59]. These toxic AMPs are secreted by eukaryotes and bacteria in order to control the number of pathogens and competitors.

Detailed studies on AMPs and similar small secretory proteins in eukaryotes indicate that their membrane targeting occurs posttranslationally and is independent of the SRP system [55]. Instead, they engage variable targeting systems like the GET system [60], Ca$^{2+}$-calmodulin [61], or Sec62 [62]. Translocation often still occurs via the Sec61 translocon, although accessory proteins like Sec62, Sec63, or BiP seem to be required [55]. Bacterial AMPs also engage variable transport systems, including a specific ABC transporter [63, 64], or the SecYEG translocon [65]. In contrast to AMPs, the targeting and insertion modes of bacterial small membrane proteins are largely unknown. Studies using conditional *secE* or *yidC* depletion strains suggested that these proteins are inserted into the membrane by a variety of different mechanisms [29]. However, depletion of either SecE or YidC is compensated by considerable proteomic changes in the respective depletion strains [66–68], which complicates data interpretation. In addition, some membrane proteins, like MtlA or TatC, can be inserted by either SecYEG or YidC [51], and therefore results obtained by conditional *secE* or *yidC* depletion strains can be misleading [31, 51]. This is exemplified by our study on YohP, which was initially suggested to insert independently of both SecY and YidC [29]. By using an in vitro transcription/translation system in combination with INV and reconstituted proteoliposomes, our data rather indicate that YohP can be inserted via a SRP-dependent mechanism by both SecYEG and YidC (Figs 3–6). This is in line with data showing that single-spanning membrane proteins and membrane proteins lacking extended periplasmic domains can be targeted by the SRP pathway to either SecYEG or YidC [4, 51].

The involvement of SRP in YohP biogenesis is rather unexpected, because the canonical model of cotranslational substrate recognition by SRP presumes a substrate length of at least 40–45 amino acids for stable SRP binding [12, 13], and thus YohP and other small membrane proteins are just too short to be recognized cotranslationally. This also highlights the problem of using tagged versions of small membrane proteins for studying their transport, because even small tags can extend their size beyond the threshold level that is required for cotranslational SRP recognition. By using nontagged or His-tagged, radiolabeled YohP, our data

demonstrate a clear posttranslational role of SRP during the insertion of YohP (Figs 4 and 5) and of YkgR (S6 Fig), another example for a small bacterial membrane protein. This is deduced from the observation that SRP maintains contact to YohP even if the ribosome is dissociated by puromycin (Fig 6 and S6 Fig) and by SRP-dependent insertion of YohP after translation has been terminated by chloramphenicol (Fig 4). A putative posttranslational role of SRP has been suggested for some eukaryotic and bacterial tail-anchored (TA) proteins [69–73]. However, at least some bacterial TA proteins, like SciP [73], contain N-terminal amphipathic helices that could serve as cotranslational recognition sites for SRP [74, 75]. A posttranslational function of SRP has been demonstrated in chloroplasts, in which cpSRP inserts a special set of nuclear encoded proteins into the thylakoid membrane [76, 77]. Here, the SRP subunits cpSRP54 and cpSRP43 form a transit complex that chaperones fully synthesized proteins to the cpFtsY-Alb3 complex at the thylakoid membrane [78, 79]. For small bacterial membrane proteins, SRP could execute a similar function, i.e., preventing their aggregation in the cytosol after they have been released from the ribosome and providing an assisted passage to the insertion sites at the membrane. However, it is important to emphasize that our data demonstrate that SRP does not execute a mere chaperone function for small membrane proteins, because YohP insertion into urea-treated membranes or proteoliposomes is only observed when both SRP and its receptor FtsY are present (Fig 3 and S8 Fig). Furthermore, YohP insertion is not supported by the targeting factor SecA or the Hsp70 chaperone DnaK (S8 Fig).

The importance of cotranslational targeting by the bacterial SRP system has been recognized in multiple studies, but its posttranslational targeting ability might be equally important for small membrane proteins and TA proteins. So far, the number of small membrane proteins and TA proteins that have been identified in *E. coli* is still comparatively low [15, 80]. However, the intergenic regions in *E. coli* alone contain more than 160,000 putative smORFs [81]. Even if only a small number of these smORFs were translated and contain a TM span, they would pose a major challenge for the cell if their targeting/insertion was unassisted.

The posttranslational role of SRP during the insertion of small membrane proteins could also include a posttargeting function. Such a role of SRP has been proposed based on in vivo experiments [82–85]. In this model, mRNAs coding for membrane proteins would be targeted to already membrane-bound ribosomes for translation and SRP would be required for the subsequent insertion [86]. This alternative model is in line with the membrane enrichment of mRNAs encoding for membrane proteins [52] and the observation that a membrane-tethered SRP is fully functional [87]. Thus, not only might the contribution of SRP for small membrane protein insertion deviate from the classical SRP model in that it acts posttranslationally, but it might also act at a posttargeting step. This model is also supported by our data, which show the membrane enrichment of the *yohP* mRNA (Fig 7). The underlying mechanism is still unknown and has to be explored further. However, the membrane enrichment of the *yohP* mRNA was observed in the absence and presence of the Shine-Dalgarno sequence, supporting a translation-independent mRNA targeting mechanism, as also observed for other mRNAs [52, 53, 88]. Furthermore, *yohP* mRNA targeting was observed in the absence of any 5′ or 3′ UTR, indicating that the coding sequence itself contains the necessary information for targeting, possibly via the strong uracil bias that has been observed in bacterial mRNAs encoding for membrane proteins [89]. The short length of the monocistronic *yohP* transcript might impair a coupled transcription-translation process [90] that is usually observed in bacteria [91] and thus favor mRNA targeting. This mechanism will likely also reduce the risk of degradation for those mRNAs that escape immediate ribosome recognition. mRNA binding to the membrane via a dedicated receptor or phospholipids would then be followed by translation and SRP-dependent posttranslational insertion into the membrane.

In summary, our data reveal a novel posttranslational function of the bacterial SRP pathway for the insertion of YohP and in all probability other small membrane proteins (Fig 8). On the one hand, SRP can bind to YohP that is released from cytosolic ribosomes and target it to its receptor FtsY that is either bound to the SecYEG translocon or to YidC. Upon contacting FtsY, SRP will release YohP, which then inserts via SecYEG or YidC into the lipid phase. The strong hydrophobicity of small membrane proteins probably favors their thermodynamic partitioning from the aqueous SecY channel into the lipid bilayer, even in the absence of any driving force provided by the ribosome as takes place during cotranslational insertion. Alternatively, SRP can also bind to YohP that was synthesized by membrane-bound ribosomes after the *yohP* mRNA has been targeted to the membrane. The mechanisms that target mRNAs to the bacterial membrane are still unknown and need to be further explored, but their existence further highlights the enormous plasticity of bacterial protein transport pathways.

## Material and methods

### Bacterial strains and plasmids

The strains and plasmids used in this study are listed in S1 Table. All small membrane proteins were cloned into the arabinose-inducible pBad24 vector [92] with a His$_6$-tag for in vivo expression experiments using the Gibson Assembly protocol [93] and the Gibson Assembly Cloning Kit (New England BioLabs) using 100 ng vector and a 1:3 molar vector/insert ratio. For in vitro and in vivo T7-dependent expression, *yohP* with and without the His$_6$-coding sequence and *ykgR* were also cloned into the pET19b (Novagen) and pRS1 [94] expression vectors. For facilitating detection of YkgR, two methionine residues were attached to the C-terminus.

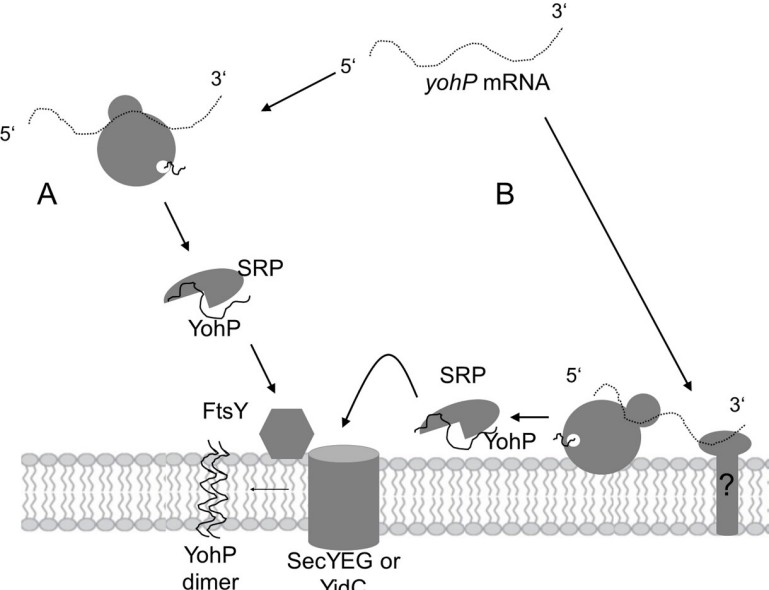

**Fig 8. Model for the posttranslational SRP-dependent insertion of the small membrane protein YohP.** (A) *yohP* mRNA is recognized by cytosolic ribosomes and translated. YohP is released from the ribosome and bound by SRP, which targets YohP to its receptor FtsY that is bound to either the SecYEG translocon or YidC. SRP dissociates from YohP upon contacting FtsY, and YohP inserts into SecY or YidC and subsequently into the lipid phase of the membrane. (B) In addition, *yohP* mRNA that is not bound by cytosolic ribosomes is targeted by a so far unknown mechanism to a membrane-bound receptor. The membrane-bound mRNA is then translated and released from the ribosome before it is bound by SRP, which then delivers YohP to SecYEG or YidC as in (A). SRP, signal recognition particle.

All PCR steps were performed with PfuUltra II fusion HS DNA polymerase (Agilent) and the provided buffers according to the manufacturer's instructions. The list of oligonucleotides for inserting TAG stop-codons, for inserting/mutating the *yohP* or *ykgR* nucleotide sequence, or for Gibson Assembly are available upon request.

## Separation of small membrane proteins by SDS-PAGE

For separation of small membrane proteins, a modified Tris-Tricine-SDS-PAGE system was used [95]. Gels were casted and gel electrophoresis was performed in a vertical dual gel system (Peqlab) with constant cooling to 3˚C. Small membrane proteins were separated by 16.5% Tricine-SDS gels with an acrylamide/bis-acrylamide 48:1.5 ratio or a 37.5:1 ratio for in vitro–expressed proteins. For simultaneous detection of cross-links or of larger proteins and small membrane proteins, a 1–2 cm spacer gel containing 10% acrylamide was added on top of the 16.5% separating gel. Gel electrophoresis was performed overnight at 4˚C and 25–27 mA. If gel drying was required for subsequent autoradiography, gels were fixed for 30–60 minutes in 35% ethanol and 15% acetic acid and washed three times with distilled water for 15 minutes each.

When whole cells were prepared for SDS-PAGE, cells were harvested and resuspended in M63 minimal medium or phosphate-buffered saline (PBS). All samples were precipitated with a final trichloroacetic acid (TCA) concentration of 5% on ice for at least 30 minutes. Subsequently, precipitated proteins were pelleted by centrifugation, and the pellet was denatured in 20 μl of Laemmli buffer containing 1% SDS maximum.

## In vivo pulse labeling and cell fractionation

For pulse-chase experiments, 10 ml LB media containing 100 μg/μl ampicillin were inoculated with C43(DE3) cells carrying pET19b-YohP or its variants and grown overnight at 37˚C. Cells were harvested and resuspended in 1 ml M63 minimal medium (20 g/L glycerol, 13.6 g/L $KH_2PO_4$, 2 g/l [$NH_4$]2$SO_4$, 0.5 mg/l $FeSO_4$ [pH 7.0] adjusted with KOH, 0.5 mg/ml thiamin, and 0.1 mM of 18 amino acids mix [all amino acids with exception of cysteine and methionine]). The resuspended preculture (200 μl) was used to inoculate 20 ml M63 medium. The cultures were incubated at 37˚C until $OD_{600}$ = 0.5–0.7, when protein expression was induced with 0.5 mM IPTG. Cells were incubated for another 20 minutes at 37˚C for allowing *T7* RNA polymerase expression. Subsequently, the endogenous *E. coli* polymerase was blocked by the addition of 50 μg/ml rifampicin and 10 minutes of incubation. Samples containing $1 \times 10^8$ cells (or $2 \times 10^8$ for cell fractionation) were collected in Eppendorf tubes. All tubes were filled up to 2 ml with M63 medium. Each sample was pulsed with 2 μl of $^{35}$S-L-methionine and $^{35}$S-L-cysteine labeling mix (11 mCi/mL, PerkinElmer) and incubated for 5 minutes at 37˚C. Afterward, the samples were chased with an excess of cold cysteine and methionine (1 μg each). Samples (100 μl each) were immediately precipitated with 10% TCA on ice for subsequent protein denaturation and SDS- PAGE analysis, or samples of 300 μl were taken and chilled on ice for subsequent cell fractionation by ultrasonic treatment (four 15-second pulses on ice). Unbroken cells were pelleted by a 15-minute centrifugation in an Eppendorf FA-45-30-11 rotor at 30,000*g* at 4˚C. The supernatant was further centrifuged to pellet bacterial membranes in a Beckmann TLA 55 rotor at 90,720*g* for 1.5 hours at 4˚C. All samples were precipitated with 10% TCA, denatured with 20 μl loading dye at 37˚C for 18 minutes and analyzed by Tricine-SDS-PAGE and phosphor imaging.

For large-scale cell fractionation, cells expressing pBad-YohP$_{His6}$ were grown to approximately $OD_{600}$ = 1.2 on LB medium, harvested, and resuspended in INV buffer (50 mM triethanolamine acetate [pH 7.5], 200 mM sucrose, 1 mM DTT). Next, the samples were lysed in the

presence of protease inhibitors by French pressing at 800 psi, and the cell debris was removed by centrifugation at 30,000$g$ for 30 minutes in an SS34 rotor. The supernatant (S30) was further centrifuged at 150,000$g$ for 2 hours in a TLA 50.2 rotor, and the pellet containing the crude bacterial membranes was dissolved in INV buffer and loaded onto a sucrose gradient and the inner membrane fraction (inverted INVs), and the OM were separated as described [96]. For carbonate extraction, INVs (20 μg protein) were treated with 0.2 M $Na_2CO_3$ (pH 11.3) for 40 minutes at 4˚C. Samples were then centrifuged for 30 minutes at 90,720$g$ in a Beckmann TLA 55 rotor. The supernatants were neutralized with glacial acetic acid and TCA precipitated. The pellet fraction was resuspended in INV buffer and centrifuged again. The pellet was then either directly resuspended in SDS-loading buffer or resuspended in INV buffer for further PK treatment (0.5 mg/ml PK for 30 minutes at 25˚C). Samples were subsequently also TCA precipitated and separated by SDS-PAGE followed by immune detection.

## DeltaVision Ultra microscopy

BL21 *E. coli* cells containing pBad24-YohP-GFP, pBad22-SecEY(YFP)G, or pACN-YchF-GFP were grown overnight in LB medium. Precultures were diluted 1:100 in fresh medium and grown to an $OD_{600}$ = 0.5 before induction by 1 mM arabinose (pBad24-YohP-GFP; pBad22-SecEY[YFP]G) or 1 mM IPTG (pACN-YchF-GFP). Cells were subsequently collected by centrifugation and resuspended in 800 μL PBS. Cell culture (5 μl) was placed under sterile conditions on an agarose-covered (1.5% agarose in PBS) glass bottom in a glass-bottom dish (35-mm dish with a 20-mm bottom well and a glass thickness of 0.16–0.19 mm, Cellvis, Mountain View, CA, USA). Plasmid-free *E. coli* strains served as a control for autofluorescence and noninduced *E. coli* strains were used for expression control.

For monitoring mRNA localization in *E. coli*, *E. coli* BL21 cells carrying pBAD24-MS2-GFP and the vector pSC.YohP-MS2.6x, containing the YohP sequence fused to the MS2-recognition sequence, but without the Shine-Dalgarno sequence, were inoculated overnight with 50 μg/mL ampicillin and 35 μg/mL chloramphenicol. The following day, a preculture was inoculated 1:100 and grown until an $OD_{600}$ = 0.5; cells were then induced with 4 mM IPTG for 40 minutes to induce mRNA expression and with 0.1 mM arabinose to induce the MS2-GFP protein for 20 minutes. Cell culture (1 mL) was centrifuged for 5 minutes at 2,300$g$ at room temperature, and the pellet was resuspended in 800 μL PBS followed by facultative Nile red staining (0.0035 g Nile red was dissolved in 1 ml acetone and diluted 1:20 in the bacterial cell suspension). The cell suspension was then incubated at 37˚C for 20 minutes. After staining, 5 μl cell culture was transferred to a clean glass-bottom dish (35-mm dish with 20-mm bottom well # 1.5 Glass [0.16–0.19 mm]) and then covered with 800 μL of PBS/1.5% agarose and a cover slip. During imaging, the background fluorescence from *E. coli* cells was also monitored by growing cells under the same conditions but without any inducer.

The same imaging conditions were used for monitoring protein and mRNA localization. Imaging was performed with a DeltaVision Ultra High Resolution Widefield Microscope (GE Healthcare, Munich, Germany) at 100× magnification with an Olympus UPlanSapo objective (Olympus, Hamburg Germany; numerical aperture 1.35) and immersion oil with a refractive index of 1.5120. Images were taken at room temperature. Exposure time was 0.2 second for GFP and at 35% laser power. DIC imaging was performed at 5% laser power and for 0.075-second exposure time. Recording, using camera sCMOS pro edge (PCO, Kelheim, Germany), was performed using a 3-μm Z-scan with optical section spacing of 0.1 μm. Acquired images were deconvolved using the Acquire Ultra software (softWoRx, GE Healthcare, Munich, Germany) and further analyzed with ImageJ/Fiji.

## Transmission electron microscopy

Primary fixation was done by 4% paraformaldehyde and 2% glutaraldehyde in 0.1 M sodium cacodylate buffer. After fixation, cells were embedded in 2% low melting agarose (Invitrogen 16520050) in 0.1 M cacodylate buffer and postfixed in 1% osmium tetroxide in ddH2O for 60 minutes on ice and then washed 6 times in ddH2O. The embedded cells were incubated in 1% aqueous uranyl acetate solution for 2 hours in the dark and washed 2 times in ddH2O. Dehydration was performed by 15-minute incubation steps in 30%, 50%, 70%, 90%, and 2× 100% ethanol and 2× 100% acetone. After embedding in Durcupan resin, ultrathin sections were performed using a UC7 Ultramicrotome (Leica) and collected on formvar-coated copper grids. Poststaining was done for 1 minute with 3% lead citrate in ddH2O and subsequent washing and drying. Grids were imaged using a Leo 912 transmission electron microscope (Zeiss).

## In vitro protein synthesis and protein purification

For protein transport assay, proteins were synthesized in vitro using a purified transcription/ translation system composed of CTFs and high salt-washed ribosomes [39]. The $^{35}$S-Methionine/$^{35}$S-Cysteine labeling mix was obtained from Perkin Elmer (Wiesbaden, Germany). For in vitro cross-linking experiments, proteins were synthesized in an S135 cell extract, which contains all cytosolic components of the *E. coli* cell [97]. INVs of *E. coli* cells were prepared as described above. U-INVs were generated by incubating INVs for 1 hour with 6 M urea in INV buffer (50 mM triethanolamine acetate [pH 7.5], 200 mM sucrose, 1 mM DTT) on ice. Subsequently, INVs were diluted 4-fold with INV buffer and centrifuged in a TLA55 rotor for 2 hours at 55,000 rpm through a 750 mM sucrose cushion in INV buffer. The pellet was resuspended in INV buffer and centrifuged again as above. After the second centrifugation, the pellet was resuspended in INV buffer and stored at −80°C. Unless otherwise stated, INVs or U-INVs were added to the in vitro transcription/translation reaction 5 minutes after start of the reaction, and the reaction mixture was incubated for an additional 25 minutes. When indicated, samples were incubated for 10 minutes at 37°C with 35 μg/ml chloramphenicol for inhibiting translation and then centrifuged for 30 minutes at 55,000 rpm in a Beckmann TLA55 rotor. One-half of the in vitro reaction was then directly precipitated with 10% TCA, and the other half was first treated with 0.5 mg/ml PK for 20 minutes at 25°C and only then TCA precipitated. PK was inactivated in 10% TCA by incubation for 10 minutes at 56°C. Next, the samples were denatured at 56°C for 10 minutes in 35 μl of TCA loading dye (prepared by mixing one part of Solution III [1 M dithiothreitol] with 4 parts of Solution II [6.33% SDS (w/ v)], 0.083 M Tris-Base, 30% glycerol, and 0.053% Bromophenol blue) and 5 parts of Solution I (0.2 M Tris, 0.02 M EDTA [pH 8]) and analyzed on SDS-PAGE by phosphor imaging.

For SecY inhibition assays, IpomF (20 mM stock in DMSO) was further diluted with DMSO to the desired concentration and subsequently incubated with INV buffer or INV for 5 minutes at 30°C. IpomF-pretreated INVs were subsequently used for in vitro transport assays. As a control, INVs were preincubated with DMSO.

Protein purification followed previously described protocols for SecYEG [87], YidC [51], SecA [51], Ffh [87], and FtsY [98]. Ffh was concentrated on a 10-kDa centrifugal filter (Amicon Ultra, Witten, Germany) and rebuffered in HT buffer + 50% Glycerol (50 mM HEPES [pH 7.6], 100 mM KOAc [pH 7.5], 10 mM Mg[OAc]$_2$, 1 mM DTT) using a PD-10 column (GE Healthcare, Munich, Germany). The protein was stored at −20°C. FtsY was rebuffered in HT buffer using a PD-10 column (GE Healthcare, Munich, Germany) and stored at −80°C. For in vitro synthesis of 4.5S RNA, pT7/T3α19, carrying the 4.5S RNA coding sequence [39] was linearized using BamHI. In vitro transcription was performed using the AmpliScribe

T7-Flash Transcription kit (Epicentre Biotechnologies, Madison, WI, USA). The 4.5S RNA was purified using the RNA purification kit (Quiagen, Hilden Germany) and stored at −80˚C. Purified DnaK was purchased from Abcam (Cambridge, UK) and dissolved in HT buffer.

*E. coli* phospholipids were purchased from Avanti polar lipids (Alabaster, AL, USA), and liposomes were generated as described [99], representing a phospholipid composition of 70% phosphatidylethanolamine, 25% phosphatidylglycerol, and 5% cardiolipin. SecYEG- and YidC proteoliposomes were created as described [51, 87]. In brief, 200 μg of liposomes and 14–16 μg of purified SecYEG or YidC were prepared in 150 μl buffer (50 mM triethanolamine acetate [TeaOAc] [pH 7.5], 1 mM DTT, and 1.5% octyl-glycoside). The samples were dialyzed with PL-buffer (50 mM TeaOAc [pH 7.5], 1 mM DTT), pelleted, and resuspended in PL-buffer to a final protein concentration of 100 ng/μl and stored at −80˚C. Before each use, proteoliposomes were briefly sonicated. In in vitro protein transport assays, 1 μl liposomes or proteoliposomes were used per in vitro reaction.

## In vivo and in vitro cross-linking

pBpa for cross-linking was obtained from Bachem (Bubendorf, Switzerland). For site-directed in vivo cross-linking, C43(DE3) cells containing both pET19b-YohP(I4pBpa)$_{His}$ and pEVol were cultured overnight in LB medium at 37˚C. The overnight culture (10 ml) was further used for inoculation of 1 l LB medium supplemented with 1 ml pBpa (final concentration 0.5 M, dissolved in 1 M NaOH), 50 μg/μl of ampicillin, and 35 μg/μl of chloramphenicol. The cultures were further incubated at 37˚C until they reached the early exponential growth phase ($OD_{600}$ = 0.5–0.8) and induced with 0.5 mM isopropyl 1-thio-β-D-galactopyranoside (IPTG) and 0.02% arabinose. After induction, the cultures were grown for 3 hours at 37˚C, cooled down on ice for 10 minutes, and harvested by centrifugation at 3,738$g$ in a JLA 9.1000 rotor for 10 minutes. The cell pellets were resuspended in 10 ml of PBS buffer (137 mM NaCl, 2.7 mM KCl, 10 mM $Na_2HPO_4$, and 1.76 mM $KH_2PO_4$) and divided in two multiwell plates. One plate was exposed to UV light on ice for 20 minutes (UV chamber: BLX-365, from Vilber Lourmat) while the other plate was kept in the dark. After UV irradiation, the cell suspension was transferred to 50-ml Falcon tubes, and cells were collected by centrifugation at 4,500$g$ for 10 minutes in an Eppendorf A-4-44 rotor. Each cell pellet was resuspended in 10 ml of resuspension buffer (50 mM Tris/HCl [pH 7.5], 300 mM NaCl, 10 mM Mg[OAc]$_2$) for subsequent YohP purification. Next, the samples were lysed in the presence of protease inhibitors by French pressing at 800 psi, and the cell debris was removed by centrifugation at 30,000$g$ for 30 minutes in a SS34 rotor. The supernatant was further centrifuged at 244,061$g$ for 1.5 hours in a TLA 50.2 rotor, and the pellet containing bacterial membranes was dissolved in 5 ml of resuspension buffer. Membranes were solubilized by the addition of 1% n-dodecyl β-D-maltoside (DDM, final concentration) for 1 hour at 4˚C on a turning wheel. For YohP purification, metal-affinity chromatography using TALON (Clontech, Mountain View, CA, USA) was used. The resin was equilibrated three times at 4˚C for 30 minutes with washing buffer (0.03% DDM, 50 mM Tris/HCl [pH 7.5], 300 mM NaCl, 10 mM Mg[OAc]$_2$, 10% glycerol [w/v], 5 mM Imidazole [pH 7.5]), before the solubilized membrane suspension was added. The material was recovered by centrifugation and washed five times for 10 minutes at 4˚C with 10 ml of washing buffer. After the fifth washing step, YohP was eluted in four steps in a total volume of 2 ml with elution buffer (0.03% DDM, 50 mM Tris/HCl [pH 7.5], 10 mM Mg[OAc]$_2$, 10% glycerol [w/v], 200 mM Imidazole [pH 7.5]). Sample purity and concentration was determined by SDS-PAGE and BCA assays (Pierce).

For site-directed in vitro photo-cross-linking, an S-135 cell extract of cells containing the pSUP plasmid was used as described before [97]. YohP(I4pBpa)$_{His}$, YohP(F27pBpa)$_{His}$, YkgR

(V6pBpa)$_{His}$, and YohP$_{His}$ as a control were translated in vitro, in the presence of additional purified SRP or INV when indicated. After synthesis, puromycin (final concentration 0.8 mM) was added and samples were incubated for 15 minutes at 25°C for releasing nascent chains. Afterward, samples were exposed to UV light for 20 minutes on ice. Proteins were precipitated with 5% TCA (final concentration) for at least 30 minutes, denatured with 20 μl loading dye at 37°C for 18 minutes, and analyzed on SDS-PAGE by phosphor imaging.

For DSS cross-linking of RNCs of the membrane protein MtlA, MtlA-RNCs of 189-amino-acid length (MtlA-189) were synthesized in vitro in the presence of an antisense oligonucleotide (5′-ACCGTGGTTAATGGCGTTGTT-3′; 60 μg/ml) as described [45, 100]. For DSS cross-linking, in vitro synthesis was performed in HEPES/NaOH buffer (pH 7.5) instead of triethanolamine acetate. Cross-linking was induced by the addition of 2.5 μl of a 25 mM DSS-solution (Pierce) prepared in DMSO, to a 25 μl in vitro reaction mixture. The sample was incubated for 30 minutes at 25°C and then quenched by Tris/HCl (pH 7.5) at a final concentration of 50 mM for 15 minutes at 25°C. Samples were then TCA precipitated as above. Puromycin treatment also followed the above described protocol.

## Immune detection and antibodies

For immune detection of small membrane proteins after SDS-PAGE, samples were electro-blotted onto PSQ 0.2-μm membranes (GE Healthcare) with 2 mA/cm$^2$ in a semidry system (transfer buffer 48 mM Tris, 39 mM Glycine, 20% methanol [v/v]). In vivo cross-linked samples were electro blotted onto Nitrocellulose 0.45-μm membranes (GE Healthcare) with 750 mA for 2 hours in a tank buffer system (transfer buffer: 50 mM Tris, 384 mM Glycine, 20% Ethanol [v/v], 0.02% SDS [w/v]). Membranes were blocked with 5% milk powder in T-TBS buffer for at least 1 hour. Polyclonal antibodies against SecA, Ffh, Trigger factor, and FtsY were generated in rabbits against the complete and SDS-denatured protein [39]. Antibodies against the SecY peptide MAKQPGLDFQSAKGGLGELKRRC were raised in rabbits by GenScript Biotech (Leiden, the Netherlands) and have been validated before [4, 87, 101]. Antibodies against YfgM were a gift of Dan. O. Daley, University Stockholm, and have been validated before [41, 94]. Monoclonal peroxidase-conjugated antibodies against the His6-tag (HisProbe-HRP Conjugate) were purchased from Thermo Scientific and from Roche. Peroxidase-coupled goat anti-rabbit antibodies (Caltag Laboratories, Burlingame, CA, USA) were used as secondary antibodies with ECL (GE Healthcare). Immune precipitation was performed on a 10-fold scaled-up in vitro reaction using polyclonal α-Ffh antibodies, covalently linked to protein A–sepharose matrix (GE Healthcare). For binding of the antibodies, 5-mg protein A–sepharose beads were equilibrated twice in 1 ml detergent buffer (150 mM NaCl, 25 mM Tris/HCl [pH 6.8], 5 mM EDTA, 1% Triton X-100, 0.5% DDM) at 4°C for 5 minutes and recovered by centrifugation at 13,000 rpm in a tabletop centrifuge. The beads were incubated with 15 μl α-Ffh antiserum at 4°C overnight. After pelleting and washing of the beads in detergent buffer, the SDS-denatured sample after cross-linking (10-fold scaled-up reaction, 250 μl) was added to the beads resuspended in detergent buffer (1 ml) and incubated for at least 1 hour at 4°C on a rotating wheel. Samples were then pelleted, washed twice with equilibration buffer (25 mM Tris/HCl [pH 6.8]), and finally resuspended in SDS-loading dye. After denaturation at 37°C for 18 minutes, the samples were separated by SDS-PAGE.

## Data quantification and statistical analyses

Western blot and autoradiography samples were analyzed by using the ImageQuant (GE Healthcare) or ImageJ/Fiji plug-in software (NIH, Bethesda, MD, USA). All experiments were performed at least twice as independent biological replicates, and representative gels/blots/

images are shown. When data were quantified, at least three independent biological replicates with several technical replicates were performed. Mean values and SEM values were determined by using either Excel (Microsoft) or GraphPad Prism (GraphPad Prism, San Diego, CA, USA).

## Supporting information

**S1 Fig. Localization of YohP and SecY in *E. coli* cells.** YohP-GFP and SecY-YFP were in vivo expressed, and imaging was performed with a DeltaVision Ultra High Resolution Widefield Microscope (GE Healthcare, Munich, Germany) at 100× magnification. Recording, using camera sCMOS pro edge (PCO, Kelheim, Germany), was performed using a 3-μm Z-scan with 0.1-μm sectioning, and the different scans of both the fluorescence channel and the merged fluorescence/bright-field picture are shown. GFP, green fluorescent protein; YFP, yellow fluorescent protein.
(PDF)

**S2 Fig. Localization of YchF in *E. coli* cells.** YchF-GFP was in vivo expressed and imaging was performed with a DeltaVision Ultra High Resolution Widefield Microscope (GE Healthcare, Munich, Germany) at 100× magnification. Recording, using camera sCMOS pro edge (PCO, Kelheim, Germany), was performed using a 3-μm Z-scan with 0.1-μm sectioning, and the different scans of both the fluorescence channel and the merged fluorescence/bright-field picture are shown. GFP, green fluorescent protein.
(PDF)

**S3 Fig. Coomassie staining of samples after cell fractionation.** *E. coli* cells carrying pBad-YohPHis were induced with 0.2% arabinose when indicated and subsequently fractionated by differential centrifugation as described in the legend to Fig 1D. Of each fraction, an aliquot corresponding to 40 μg protein was separated by SDS-PAGE and stained with Coomassie blue. YohP$_{His}$, His-tagged YohP.
(PDF)

**S4 Fig. YohP dimerizes via an unusual glycine motif.** (A) YohP or the YohP(G15A/G21A) mutant were in vivo expressed and pulse-labeled. Whole cells were then TCA precipitated after the indicated time points, separated by SDS-PAGE and analyzed by autoradiography. ± refers to a nonspecifically labeled band. (B) As in (A), but cells were fractionated after cell breakage into an S30 (cytosol and membrane), S150 (cytosol), and P150 (membrane) fraction, as described in the legend to Fig 1. (C) YohP, the YohP(G15A/G21A) mutant, or the two single mutants YohP(G15A) and YohP(G21A) were in vivo expressed, pulse-labeled, and analyzed by autoradiography as above. TCA, trichloroacetic acid.
(PDF)

**S5 Fig. U-INVs lack SecA, Ffh, and FtsY.** (A) Urea-treated vesicles were decorated after SDS-PAGE and western blotting with antibodies FtsY, Ffh (upper panel), or SecA (lower panel). (B) The translocation of OmpA into U-INV requires the presence of SecA. Translocation of OmpA was analyzed as described in Fig 3. (C) The integration of MtlA into U-INV requires the presence of FtsY and SRP. MtlA integration was analyzed as described in Fig 3. Underlying data for this figure can be found in S1 Data. INV, inner membrane vesicle; MtlA, mannitol permease; OE, INV from an SecYEG-overexpressing strain; SRP, signal recognition particle; U-INV, urea-treated INV.
(PDF)

**S6 Fig. Co- and posttranslational interaction of SRP with membrane protein substrates.**
(A) MtlA-189 RNCs (189 amino acids) of the inner membrane protein MtlA were in vitro synthesized using an antisense oligonucleotide approach [1]. When indicated, purified SRP was added to MtlA-RNCs. Cross-linking was induced by the addition of DSS, an amine-reactive cross-linker that had been used before for monitoring MtlA-SRP contacts [1, 2]. When indicated, puromycin (1 mM) was added prior to DSS cross-linking. Note that full-length MtlA is also visible, because the antisense oligonucleotide approach for generating RNCs does not allow for a complete suppression of full-length protein synthesis. The identity of the Ffh-MtlA-189 RNC cross-link has been validated in multiple studies [1–3]. (B) YkgR(V6pBpa)$_{His}$ was in vitro synthesized and, when indicated, incubated with purified SRP and UV-exposed in the presence of INV or after puromycin treatment. Samples in (A) and (B) were subsequently TCA precipitated, separated by SDS-PAGE, and visualized by autoradiography. At least two biological replicates were performed, and a representative gel is shown. DSS, disuccinimidyl suberate; MtlA, mannitol permease; RNC, ribosome-associated nascent chain; SRP, signal recognition particle; TCA, trichloroacetic acid; YkgR(V6pBpa)$_{His}$, YkgR containing the UV-reactive cross-linker para-benzoyl-L-phenylalanine at position 6.
(PDF)

**S7 Fig. IpomF inhibits the membrane insertion of YohP.** MtlA and YohP were in vitro synthesized, and membrane insertion was analyzed into INVs that were pretreated with different concentrations of the inhibitor IpomF or with DMSO as control (0 mM IpomF). Indicated is the MtlA-MPF, which results from degradation of the approximately 30-kDa cytoplasmic domain by proteinase K and the membrane-protected fragment of YohP (#). Quantification of several independent experiments is shown in Fig 6B. INV, inner membrane vesicle; IpomF, Ipomoeassin F; MtlA, mannitol permease; MtlA-MPF, membrane-protected fragment of MtlA.
(PDF)

**S8 Fig. YohP does not spontaneously insert into membranes.** (A) YohP was in vitro synthesized in the absence of membranes, translation was stopped by the addition of chloramphenicol, and the sample was centrifuged for removing aggregates. The supernatant was subsequently incubated with INV or liposomes in the presence or absence of purified SRP/FtsY (20 ng/μl, each). Samples were then subjected to proteinase K treatment as before. Quantification was performed on three independent experiments, and the SEM is shown. (B) As in (A), but in vitro–synthesized YohP was incubated with SecYEG proteoliposomes in the presence of the indicated targeting factors/chaperones, which were present at a final concentration of 20 ng/μl together with a nucleotide mix (50 μM of each ATP and GTP in INV buffer). Quantification was performed on at least three independent experiments, and the SEM is shown. Underlying data for this figure can be found in S1 Data. INV, inner vesicle membrane; SEM, standard error of the mean; SRP, signal recognition particle.
(PDF)

**S1 Table. *E. coli* strains and plasmids used in this study.** The *E. coli* strains and plasmids used in this study, their genotypes, antibiotic resistance, application, and references are listed.
(XLSX)

**S1 Data. Underlying numerical data and statistical analyses.** Quantification and statistical analyses of YohP localization by western blot analyses and of YohP membrane integration by autoradiography.
(XLSX)

## Acknowledgments

We thank Narcis-Adrian Petriman for his help in setting up the cross-linking experiments for small membrane proteins and Dan O. Daley (Univ. Stockholm) for antibodies against YfgM.

This work reflects only the authors' view, and the European Union's Horizon 2020 research and innovation program is not responsible for any use that may be made of the information it contains.

## Author Contributions

**Conceptualization:** Ruth Steinberg, Andrea Origi, Ana Natriashvili, Maximilian H. Ulbrich, Hans-Georg Koch.

**Data curation:** Ruth Steinberg, Hans-Georg Koch.

**Formal analysis:** Ana Natriashvili, Joen Luirink, Wei. Q. Shi, Maximilian H. Ulbrich, Hans-Georg Koch.

**Funding acquisition:** Claudine Kraft, Wei. Q. Shi, Hans-Georg Koch.

**Investigation:** Ana Natriashvili, Pinku Sarmah, Mariya Licheva, Martin Helmstädter, Maximilian H. Ulbrich.

**Methodology:** Ruth Steinberg, Andrea Origi, Ana Natriashvili, Pinku Sarmah, Mariya Licheva, Martin Helmstädter, Maximilian H. Ulbrich.

**Project administration:** Hans-Georg Koch.

**Resources:** Princess M. Walker, Stephen High, Joen Luirink, Wei. Q. Shi.

**Supervision:** Claudine Kraft, Hans-Georg Koch.

**Validation:** Ruth Steinberg, Andrea Origi, Ana Natriashvili, Pinku Sarmah, Maximilian H. Ulbrich.

**Writing – original draft:** Ruth Steinberg, Hans-Georg Koch.

**Writing – review & editing:** Ruth Steinberg, Andrea Origi, Ana Natriashvili, Pinku Sarmah, Mariya Licheva, Princess M. Walker, Claudine Kraft, Stephen High, Joen Luirink, Wei. Q. Shi, Maximilian H. Ulbrich, Hans-Georg Koch.

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
