## [Editor Report · Decision Letter 0]

7 May 2020

Dear Dr Koch, 

Thank you for submitting your revised manuscript entitled "Post-translational insertion of small membrane proteins by the bacterial signal recognition particle" for consideration as a Research Article by PLOS Biology.

Your revision has now been evaluated by the PLOS Biology editorial staff, as well as by the original Academic Editor, and I am writing to let you know that we would like to send your submission out for external peer review.

Please re-submit your manuscript within two working days, i.e. by May 11 2020 11:59PM.

Kind regards,

Gabriel Gasque, Ph.D.,

Senior Editor

PLOS Biology

---

## [Decision Letter · Decision Letter 1]

11 Jun 2020

Dear Dr Koch,

Thank you very much for submitting a revised version of your manuscript "Post-translational insertion of small membrane proteins by the bacterial signal recognition particle" for consideration as a Research Article at PLOS Biology. This revised version of your manuscript has been evaluated by the PLOS Biology editors and by the original Academic Editor and reviewers 1, 2, and 4.

In light of the reviews (below), we are positive about your study and pleased to offer you the opportunity to address the lingering points raised by the reviewer. As you will see, reviewer 1 thinks that additional in vivo data could resolve standing ambiguities in your study. Together with the Academic Editor, we encourage you to seriously consider this request. Although we will not make addressing it experimentally a prerequisite for publication, adding these additional data seems relatively straightforward and would significantly strengthen the robustness and in vivo relevance of the work. Please also address the outstanding minor concerns of reviewers 1 and 2. 

We cannot make any decision about publication until we have seen the revised manuscript and your response to the reviewers' comments. Your revised manuscript is also likely to be sent for further evaluation by the reviewers.

We expect to receive your revised manuscript within 2 months. 

**IMPORTANT - SUBMITTING YOUR REVISION**

*Re-submission Checklist*

*Published Peer Review*

*PLOS Data Policy*

*Blot and Gel Data Policy*

Sincerely,

Gabriel Gasque, Ph.D., 

Senior Editor

PLOS Biology

REVIEWS:

Reviewer #1: The authors have addressed most of my comments during the last round of review. In general, biochemical data are much stronger and provide strong evidence that YohP can be post-translationally targeted by SRP/FtsY. Additional reconstitutions using proteoliposomes further pointed to SecYEG or YidC as potential translocases for YohP, which is a nice addition. Overall the paper is much stronger. My major remaining concern is still the lack of in vivo data to show that YohP insertion is SRP/FtsY dependent and post-translational. The additional in vivo experiment showing the membrane localization of the YohP mRNA further introduces ambiguity to the interpretation of the targeting route in vivo. Since the authors already have an in vivo assay for the membrane localization of YohP by pulse chase, it seems within their reach to at least show that YohP insertion in vivo is SRP/FtsY dependent.

Additional minor comments:

The topology of YohP via proteinase K is opposite of the conclusion in the previous draft using a different method. Given that YohP is small, could its topology be influenced by the His6 tag? I note that the authors have already shown good evidence that YohP is membrane-localized / inserted, so the topology information is not necessary in this paper and may be left for a different characterization given the current ambiguity.

Since OmpA translocation is SecA and SecYEG-dependent, is it possible that the effect of IpomF on OmpA translocation could be due to inhibition of SecA, rather than SecYEG? 

Reviewer #2: The authors have made a commendable effort to address the reviewers' comments.

1. However, many of their statements are still too definitive (for example in the abstract). The authors should simply state their observations for the constructs they examined and the assays they carried out. As just a few examples, I suggest the following changes (these are just examples, modifications need to be made throughout the manuscript):

--line 194: "…these data show that the vast majority of YohP-His is oriented in a Cin-Nout topology, …" 

--line 223: "…does not efficiently occur spontaneously in our assay, suggesting that insertion requires…"

--line 267: "…strongly stimulated YohP-His insertion." 

--lines 298-299: "In order to further test whether…"

--line 366: "…whereas our data indicate that the…"

--line 370: "To obtain further information about the insertion mode…"

Small proteins are tricky to study and as found by the authors, different assays can lead to completely different conclusions. In the previous draft, the authors concluded, based on AMS labeling, that YohP has a Nin-Cout topology. Now the authors conclude the protein has a Cin-Nout topology. Both assays have limitations. The authors should avoid stating that they have "proof".

2. Some non-standard phrasing that should be changed:

--line 203: "allows to determine"

--line 233: "for preventing possible saturation"

--line 283: "10-fold up-scaled"

--line 326: "were UV-exposed"

--line 329: "decorated with a-Ffh antibodies"

Reviewer #4: I find that the authors have done a formidable job in addressing the reviewers comments and now present a much stronger case for they study with more controls and clarification of issues that were confusing or not directly justified by the available data.

I therefore fully support its publication.

---

## [Editor Report · Decision Letter 2]

13 Aug 2020

Dear Dr Koch,

Thank you for submitting your revised Research Article entitled "Post-translational insertion of small membrane proteins by the bacterial signal recognition particle" for publication in PLOS Biology. I have now discussed this version with the Academic Editor, and we're delighted to let you know that we're now editorially satisfied with your manuscript. 

However before we can formally accept your paper and consider it "in press", we also need to ensure that your article conforms to our guidelines. A member of our team will be in touch shortly with a set of requests. As we can't proceed until these requirements are met, your swift response will help prevent delays to publication. Please also make sure to address the data and other policy-related requests noted at the end of this email.

*Copyediting*

*Published Peer Review History*

*Early Version*

*Submitting Your Revision*

Sincerely,

Gabriel Gasque, Ph.D.,

Senior Editor,

ggasque@plos.org,

PLOS Biology

DATA AVAILABILITY:

Please rename your "Quantification" file as S1 Data, making sure it has a legend and is mentioned in each figure legend.

---

## [Editor Report · Decision Letter 3]

2 Sep 2020

Dear Dr Koch,

On behalf of my colleagues and the Academic Editor, Frederick M Hughson, I am pleased to inform you that we will be delighted to publish your Research Article in PLOS Biology. 

Early Version

PRESS 

Kind regards,

Alice Musson

Publishing Editor, 

PLOS Biology

on behalf of

Gabriel Gasque,

Senior Editor

PLOS Biology